# Let Them Talk: Audio-Driven Multi-Person Conversational Video Generation

**Zhe Kong**[1,2,3]*, **Feng Gao**[2]*, **Yong Zhang**[2]†, **Zhuoliang Kang**[2], **Xiaoming Wei**[2],
**Xunliang Cai**[2], **Guanying Chen**[1], **Wenhan Luo**[3]†

[1]Shenzhen Campus of Sun Yat-sen University     [2]Meituan
[3]Division of AMC and Department of ECE, HKUST

`https://meigen-ai.github.io/multi-talk/`

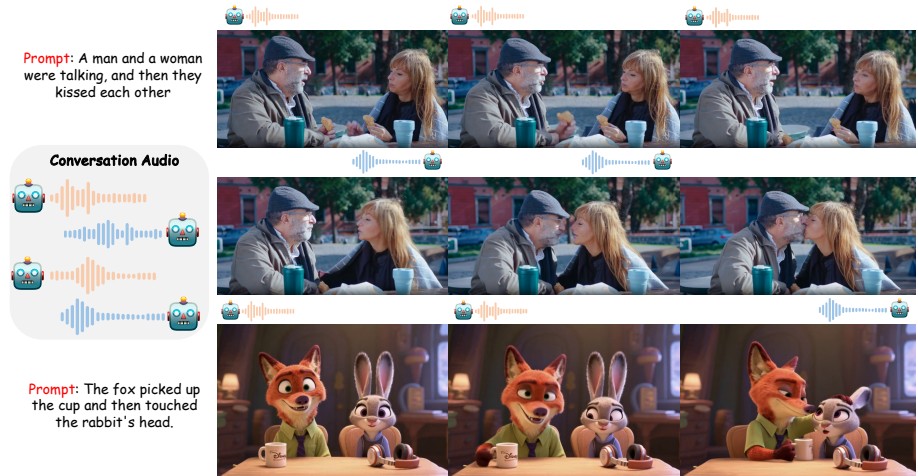

Figure 1: We propose MultiTalk, a novel framework for audio-driven multi-person conversational video generation. Given a multi-stream audio input and a prompt, MultiTalk generates a video containing interactions following the prompt, with consistent lip motions aligned with the audio.

## Abstract

Audio-driven human animation methods, such as talking head and talking body generation, have made remarkable progress in generating synchronized facial movements and appealing visual quality videos. However, existing methods primarily focus on single human animation and struggle with multi-stream audio inputs, facing incorrect binding problems between audio and persons. Additionally, they exhibit limitations in instruction-following capabilities. To solve this problem, in this paper, we propose a novel task: Multi-Person Conversational Video Generation, and introduce a new framework, MultiTalk, to address the challenges during multi-person generation. Specifically, for audio injection, we investigate several schemes and propose the Label Rotary Position Embedding (L-RoPE) method to resolve the audio and person binding problem. Furthermore, during training, we observe that partial parameter training and multi-task training are crucial for preserving the instruction-following ability of the base model. MultiTalk achieves superior performance compared to other methods on several datasets, including talking head, talking body, and multi-person datasets, demonstrating the powerful generation capabilities of our approach.

---

*Equal Contribution.

†Corresponding Author.

39th Conference on Neural Information Processing Systems (NeurIPS 2025).

# 1 Introduction

Audio-driven human animation aims to generate natural and vivid human-centric videos with synchronized facial expressions and body movements from audio control signals. This field has made significant progress recently, and existing methods can be roughly divided into two categories: talking head generation and talking body generation.

Most human animation methods [1, 2, 3, 4, 5, 6] focus on talking head generation. These methods utilize diffusion models to match audio features to visual frames, enabling the synthesis of vivid talking head videos with enhanced video quality and realistic facial expressions. However, they are constrained to achieve precise audio-aligned facial movements and often neglect other related motions, such as hand and body. Recently, several methods [7, 8, 9, 10, 11] have utilized video diffusion models [12, 13, 14] and successfully achieved talking body generation. By leveraging mixed data training strategies or using additional hand pose data, they can synchronize body movements with the audio. Despite these advancements, several constraints remain. Existing methods primarily target single-person animation and cannot handle multi-person scenarios, such as conversational video generation. They lack the capability for dual-stream audio injection. Additionally, they exhibit limitations in instruction-following capabilities. For instance, generated videos may fail to precisely follow instructions when a text prompt describes a large range of body movement.

In this paper, we propose a new task: audio-driven multi-person conversational video generation. This task has diverse applications, including multi-character movie scenes making and e-retailers' livestreaming. Compared to audio-driven single-human animation, this task presents three main challenges: 1) As conversations involve audio from multiple persons, the model should accommodate multi-stream audio inputs; 2) Each person within the conversation should be driven by only one audio stream to prevent incorrect face and audio binding; 3) Each person in the generated video is dynamic, requiring an adaptive method for person localization. Despite the success of existing methods in achieving subtle expressions and realistic motions for a single person, challenges remain in creating multiple-person videos. Specifically, existing methods cannot handle multi-stream input audio and are limited to a single audio stream. Additionally, when reference images contain multiple people, the audio tends to drive all individuals to speak simultaneously, resulting in consistent lip motions across all persons. This complicates the achievement of alternating speech in conversational video.

To complete this new task, we propose a novel framework, MultiTalk, for audio-driven multi-person conversational video generation. Multi-stream audio injection often encounters incorrect binding between the audio and the person. We investigate several schemes for audio injection and introduce the Label Rotary Position Embedding (L-RoPE) method. By assigning identical labels to audio embeddings and video latents, it effectively activates specific regions within the audio cross-attention map, thereby resolving incorrect binding issues. Furthermore, we explore a set of training strategies, including multi-stage training, partial parameter training, and multi-task training. Our observations highlight the importance of the latter two strategies. After incorporating a multi-event dataset for image-to-video, the instruction-following ability of the base model is preserved.

Our main contributions are summarized as follows: (1) We propose a novel task, *i.e.,* audio-driven multi-person conversational video generation, and introduce a novel framework to address the challenges. (2) We investigate several schemes for multi-stream audio injection and propose the Label Rotary Position Embedding method to resolve the inaccurate audio binding problem in multi-person video generation. (3) We explore a set of training strategies, including multi-stage training, partial parameter training, and multi-task training. We observe that the latter two are crucial for preserving the instruction-following ability of the base model, especially with limited compute resources and data. The multi-event dataset for the image-to-video is quite crucial. (4) We conduct evaluations on various datasets, such as talking face, talking body, and multi-person conversation. The results demonstrate the effectiveness of the proposed method.

# 2 Related Work

## 2.1 Audio-driven Human Animation

Pioneering audio-driven human animation works [15, 16, 17, 18, 19, 20, 21] typically consist of two components. They first employ an audio-to-motion model to transform motion signals into intermediate representations such as 3DMM [22] and FLAME [23]. Subsequently, motion-to-video

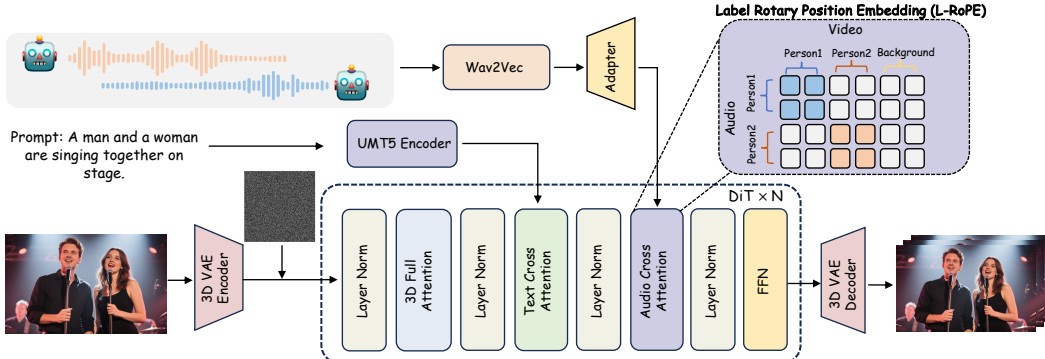

Figure 2: The overall pipeline of the proposed MultiTalk framework. Our framework incorporates an additional audio cross-attention layer to support audio conditions. To achieve multi-person conversational video generation, we propose a Label Rotary Position Embedding (L-RoPE) for multi-stream audio injection.

rendering techniques, such as GANs, are employed to project these intermediate representations into dynamic portrait animations. Despite notable successes, limitations in audio-to-motion models' ability to capture intricate facial expressions and head movements significantly constrain the authenticity and naturalness of synthesized videos.

Recently, end-to-end audio-to-video synthesis methods [1, 24, 2, 4, 3, 25, 5, 6] omit intermediate representation and directly utilize a single diffusion model to integrate audio cues with facial dynamics. These methods demonstrate enhanced potential, exhibiting superior naturalness and consistent portrait animation capability. However, they are constrained to support only head movement. To achieve audio-driven body animation, CyberHost [7] proposes a one-stage audio-driven talking body generation framework equipped with a Region Attention Module and Human-Prior-Guided Conditions to address common synthesis degradations in half-body animation. EMO2 [9] introduces a two-stage framework, first generating hand movements and subsequently using them as control signals in the second stage to enable holistic facial expressions and upper body motions. OmniHuman [8] employs a mixed data training strategy with multimodal motion conditioning to overcome the scarcity of high-quality data. EchomimicV2 [10] proposes an Audio-Pose Dynamic Harmonization strategy, requiring an additional hand pose sequence as input alongside audio. However, these audio-driven human animations can only animate a single person and cannot achieve multi-stream audio-driven image animation.

## 2.2 Video Diffusion Model

The success of text-to-image diffusion models and their downstream applications [26, 27, 28, 29] has sparked considerable interest in exploring their potential for video generation. Video diffusion models can be roughly divided into two categories: text-to-video models and image-to-video models. Early video diffusion models [30, 31, 12] typically leverage the U-Net architecture for video generation, attempting to extend the 2D U-Net pretrained on text-to-image tasks into 3D to generate continuous video frames. Recent works [32, 33, 14] have adopted a DiT (Diffusion-in-Transformer) architecture [34], significantly advancing video generation technology. These DiT-based methods replace the U-Net with a Transformer, incorporating a 3D VAE as the encoder and decoder. By expanding the training dataset, DiT networks learn motion priors for various objects and scenes. Video diffusion models demonstrate substantial potential in tackling intricate video generation tasks and provide a strong visual backbone for various downstream tasks [35, 36, 37, 38, 39]. Due to its excellent performance in human generation, a DiT-based image-to-video diffusion model is adopted as the backbone of our method to fully leverage its human generative prior.

## 3 Method

The overall architecture of the proposed method is illustrated in Fig. 2, showcasing an audio-driven multi-person conversational video generation framework. In Section 3.1, we first briefly describe the network architecture of the video foundational model. Then, in Section 3.2, we introduce the

integration of audio conditions via an audio cross-attention mechanism for single-person animation. Subsequently, in Section 3.3, we present our investigation into multi-stream audio injection and introduce the proposed L-RoPE method for audio and person binding. In Section 3.4, we explain our training strategy. Finally, we describe our method for long video generation in Section 3.5.

## 3.1 Preliminaries

In this study, we adopt a DiT-based video diffusion model as our foundational model, which is built upon the DiT architecture and incorporates a 3D Variational Autoencoder (VAE). This design achieves compression in both spatial and temporal dimensions. A textual encoder is utilized to generate the text-conditioned input, denoted as $c_{text}$. Additionally, the extracted global context from the CLIP image encoder [40] is injected into the DiT model along with $c_{text}$ via decoupled cross-attention.

## 3.2 Audio-Driven Single Person Animation

Our foundational model is an image-to-video diffusion model capable of animating a reference image to generate a video. However, it does not natively support audio as an input. To incorporate an additional audio condition, we add layers consisting of layer normalization and an audio cross-attention mechanism after the text cross-attention in each DiT block.

**Audio Embedding Extraction**    To extract acoustic audio embeddings, we employ Wav2Vec [41], a widely utilized audio feature extractor. In audio-driven human animation, since current motion is influenced by both preceding and succeeding audio frames, we follow [1] and concatenate audio embeddings proximal to the current frames, described as follows:

$$a_i = Concat(a_{i-\lfloor \frac{k}{2} \rfloor}, \cdots, a_i, \cdots, a_{i+\lfloor \frac{k}{2} \rfloor}) \tag{1}$$

where $k$ denotes the context length.

In the audio cross-attention layer, queries are derived from video latents, while keys and values originate from audio embeddings. These elements execute frame-by-frame attention calculations. Due to the temporal compression of the 3D VAE, the frame length of video latents is shorter than that of audio embeddings, complicating direct calculations between them. To address this, we propose an audio adapter for audio compression. Specifically, suppose the input audio contains $l$ frames. We first divide the audio embedding into the initial frame $a_1$ and the subsequent frames $a_{[2:l]}$ along the temporal dimension. Next, we downsample $a_{[2:l]}$ get $Down(a_{[2:l]})$, and then encode $a_1$ with $Down(a_{[2:l]})$ separately through several MLP layers. After concatenating, we encode the concatenated features to obtain the compressed audio condition $c_a$. This process is represented as:

$$c_a = MLP(Concat(MLP(a_1), MLP(Down(a_{[2:l]})))). \tag{2}$$

## 3.3 Audio-Driven Multi-Person Animation

Existing methods fail to address the problem of multi-human generation driven by multi-audio streams. In this paper, we introduce a novel task: audio-driven multi-person conversational video generation. To tackle this challenge, we propose a new framework, MultiTalk, specifically designed to handle multi-stream audio injection and rectify incorrect audio and person binding. The overall architecture of MultiTalk is depicted in Fig.2. We first investigate several schemes for multi-stream audio injection. Then, to accurately identify each person's motion region in generated videos, we propose an adaptive person localization method. Finally, we introduce the proposed L-RoPE method to effectively bind audio and persons.

**Multi-stream Audio Injection Schemes.**    Multi-person conversational video generation, unlike single audio-driven video generation, requires the model to accommodate multi-stream audio inputs. To find an effective method for audio injection, we explore four distinct injection schemes, as illustrated in Fig. 3.

Our first attempt involved directly concatenating the multi-stream audio embeddings $z_{a1}$ and $z_{a2}$, then calculating the audio cross-attention results with video latent $z_t$, as shown in Fig. 3 a). Another strategy is to calculate the multi-stream audio embeddings $z_{a1}$ and $z_{a2}$ separately with $z_t$, and then followed by an adding operation to calculate these two components, as seen in Fig.3 b). However,

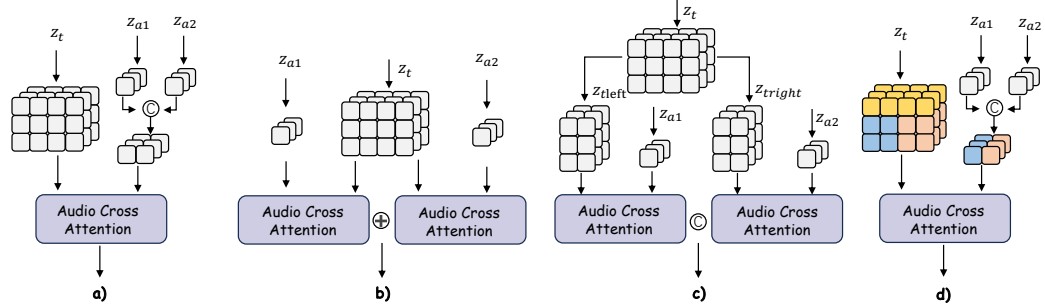

Figure 3: Investigation on different injection strategies for multi-stream audio condition.

these two attempts failed to bind the multi-stream audio with its corresponding video latent region. The network cannot learn to bind audio to different persons through training directly. Given that the individuals in the generated video are typically positioned on the left and right sides, we attempted to simplify binding by splitting the video latents into left and right segments, as demonstrated in Fig. 3c). Each video latent segment computes attention results with the corresponding audio embedding separately, and the two attention results are concatenated as the final output. Although this attempt successfully binds multi-stream audio to different persons, its generalization capacity is limited. Specifically, it is only effective for videos with minimal movement range. When a person exhibits extensive motion, directly applying this simple operation results in audio binding failures. To address these shortcomings, we propose an adaptive method for multi-stream audio injection, named L-RoPE, as illustrated in Fig. 3d).

**Adaptive Person Localization.** Before utilizing L-RoPE, the model must adaptively track the localization of each individual. Given a reference image $I$ contains two persons, we first find the subject localization within $I$, resulting in the set $M = \{M_{p1}, M_{p2}, M_b\}$. Here, $M_{p1}$ and $M_{p2}$ represent the mask regions for each person, and $M_b$ denotes the mask covering the background in the reference image. Collectively, they satisfy the relation $I = M_{h1} \cup M_{h2} \cup M_b$. The self-attention map reflects the similarity of generated video latents across different frames. In the I2V model, the first frame of the video also serves as the reference image, enabling the creation of a reference-image-to-video attention map $A_{r2v} \in R^{fhw \times 1hw}$, as depicted in Fig. 4 a). Here, $f$ denotes the frame length in latent space, while $h$ and $w$ represent the height and width, respectively. Since the reference image contains multiple subjects within $M$, we calculate the average similarity of each latent in $z_t$ with the subjects in the reference image, yielding $S \in R^{fhw \times 3}$. In this matrix, $S(i, j)$ represents the similarity between the $i$-th token in the video latents and the $j$-th subject in $M$. By leveraging the similarity captured in the self-attention map, we can adaptively locate each person in the video.

**L-RoPE for Audio and Person Binding.** Rotary Position Embedding (RoPE) [42] is a relative positional encoding technique that effectively captures inter-token relationships in large language models (LLMs). Known for its proficiency in modeling long sequences, RoPE has also been employed in video diffusion models, such as CogVideoX [32], Hunyuan Video [33], and Wan [14], among others, to facilitate multi-resolution, multi-aspect ratio, and variable duration video generation. It is utilized to generate position-aware query and key embeddings for time, height, and width within the video latents during the self-attention layer of the DiT block. In this paper, we introduce the Label Rotary Position Embedding (L-RoPE) method, aimed at binding multi-stream audio to multiple persons within the audio cross-attention layers of the DiT block.

Specifically, take the query $q$ as an example. $q$ is a sequence of $N$ vectors $\{q_i\}_{i=1}^N$. We compute an angle $\theta_i$ for each vector $q_i$ using its label $l_i \in \mathbb{R}$, and rotate $q_i$ with $\theta_i$ to obtain $\hat{q}_i$:

$$\theta_i = l_i * \theta_{base} \tag{3}$$

$$\hat{q}_i = LRoPE(q_i, l_i) = q_i e^{l_i \theta_i} \tag{4}$$

where $\theta_{base}$ is a pre-defined base angle.

In the audio cross-attention mechanism, queries are derived from the video latent $z_t$, whereas keys and values originate from the multi-stream audio embeddings $z_{a1}$ and $z_{a2}$. Appropriately assigning

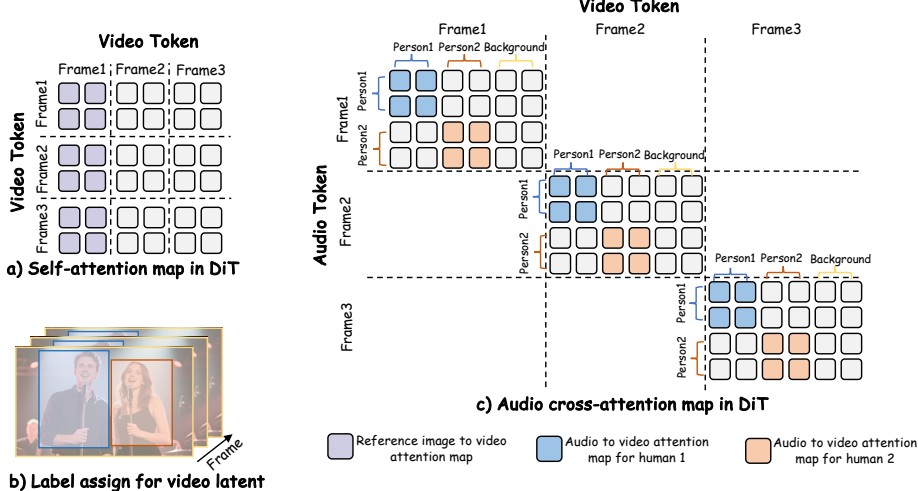

Figure 4: Analysis for different components in the DiT. a) We utilize the reference-image-to-video self-attention map in DiT for person localization. b) We assign different labels to the multiple subjects in the video. c) Assigning a close label for video and audio can activate a specific region in the audio cross-attention map.

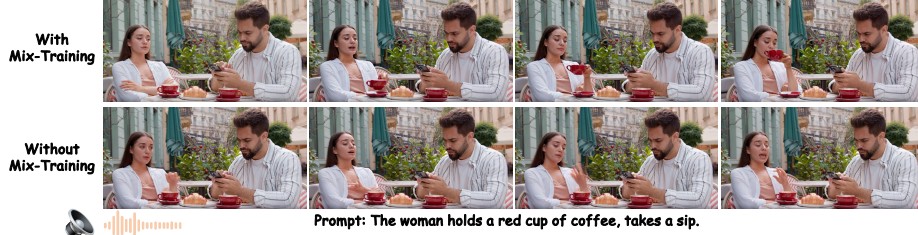

Figure 5: Instruction-following capability comparison between different training strategies.

labels $l$ to video and multi-stream audio is crucial. As depicted in Fig. 4b, video latents encompass regions corresponding to multiple persons and the background. We adopt a specific strategy for label assignment. For person regions, due to varying sensitivity driven by audio in different parts of the body, we first assign a numerical range for each person, $(a, b)$. Then, we determine the category $C \in \mathbb{R}^{fhw}$ of each vector in $q$ through $argmax_j(S[i,j])_{i=1}^{fhw}$ . Finally, taking the first person as an example, the label for person1 can be calculated through the normalization function, $Norm(S[i,j]_{j=C[person1]}, a, b) = \frac{s_{i,j} - min(S_{,j})}{max(S_{,j}) - min(S_{,j})} * (b - a) + a$. This method is applied for each person in the same manner, but using different label ranges. Specifically, we define the visual label range as $\{0 - 4\}$ for the first person and $\{20 - 24\}$ for the second person. Conversely, for the background and dual audio, they directly utilize a static value as their label. The background should not be associated with audio, hence we assign it the label 12. For multi-audio embedding, as shown in Fig. 3d, we first concatenate the multi-stream audio embeddings and subsequently assign different labels $c_{a1}$ and $c_{a2}$ to them. To bind the multi-stream audio with the two persons respectively, we set $c_{a1}$ as 2 and $c_{a2}$ as 22.

## 3.4 Training Strategy

**Two-stage training.** The training stages and associated data sources are essential for achieving effective multi-person animation. We divide the training process into two stages, progressively enhancing the model's capabilities in audio and lip synchronization. The first stage primarily focuses on developing the model's ability to animate a single person. Subsequently, in the second stage, we employ training data that contains dual-stream audio to facilitate multi-human animation.

**Partial Parameter Training.** In our method, only the network parameters in the audio cross-attention and audio adapter are updated, while all other network parameters are frozen during training. We also compare this strategy with full parameter training. Our findings indicate that network training parameters are crucial; when the compute resources and data are limited, fully parameterized training can lead to not only the degradation in the model's instruction-following ability, especially for motion and interaction, but also cause hand and object distortion. Conversely, training only the audio cross-attention does not result in this issue and the instruction-following ability of the base model can be well preserved.

**Multi-task training.** During training, we adopt a multi-task hybrid paradigm, dividing model training into multiple tasks, including audio + image to video (AI2V) training and image to video (I2V) training. Different tasks utilize distinct training data while sharing the same network parameters. For AI2V tasks, both the reference image and audio are used as conditions. In the I2V task, the audio condition is removed by zeroing the audio embedding. Additionally, the training data used for the I2V task is unique, comprising mainly of multi-event videos with interactions among human, object, and scene, which is crucial for the alignment between the motion description in the prompt and the generated video.

Multi-task training substantially impacts the results, as shown in Fig.5. Utilizing only talking head and talking body data for AI2V training diminishes the network's instruction-following capability. Conversely, incorporating I2V training allows the model to retain its instruction-following ability.

## 3.5 Long Video Generation

Although the model can generate video lengths of up to a few frames, this is still insufficient for real-world applications. To address this issue, we introduce an autoregressive-based method to facilitate long video inference. Specifically, within the I2V model, the first frame of the video is typically used as the condition for inference. In contrast, we incorporate the last $5$ frames of the previously generated video as additional conditions for inference. Following 3D VAE compression, these conditional frames are reduced to 2 frames of latent noise. We pad zeros to the subsequent frames and concatenate them with latent noise and a video mask. These are then input into DiT for inference, enabling longer video generation.

# 4 Experiments

## 4.1 Settings

**Datasets.** We collect a video dataset of about 2K hours for the first stage training, which covers the face or body of a single talking person. We also collect about 200K video clips that contain multiple events and human-object/environment interactions. The average clip duration is about 10 seconds. For the second stage training, we collect 100 hours of videos consisting of conversations between two persons. For evaluation, we employ three distinct types of testing datasets: the talking head dataset, the talking body dataset, and the dual-human talking body dataset with interactive scenarios. For the talking head dataset, we employ two publicly available datasets, HDTF [43], and CelebV-HQ [44] for evaluation purposes. For the talking body dataset, we utilize the EMTD [10] dataset. Since we are the first to propose a dual-human talking body task, no public dataset is available. We collect a dataset containing $40$ videos (referred to as MTHM) sourced from the internet.

**Evaluation Metrics.** We utilize the commonly used metrics to evaluate the methods. Frechet Inception Distance (FID) [45] and Fréchet Video Distance (FVD) [46] are used to assess the quality of the generated data. Expression-FID (E-FID) is used to evaluate the expressiveness of the facial in the generated video. Sync-C [47] and Sync-D [47] are utilized to measure the synchronization between audio and lip movements.

**Implementation Details.** We adopted Wan2.1-I2V-14B as the foundational video diffusion model for our experiments. The model is trained using a constant learning rate of $2e-5$, incorporating a warm-up strategy, and optimized using the AdamW optimizer. During training, we only fine-tuned the audio cross-attention layer and adapter while keeping other layers frozen. The proposed method was trained using $64$ NVIDIA H800-80G GPUs. In stage $1$ of the training process, the batch size was set to $64$, whereas in stage $2$, the batch size was adjusted to $32$.

Table 1: Quantitative comparison with other competing methods on talking head generation, including HDTF and CelebV-HQ datasets.

| Methods | HDTF | | | | | CelebV-HQ | | | | |
|---|---|---|---|---|---|---|---|---|---|---|
| | Sync-C↑ | Sync-D↓ | E-FID↓ | FID↓ | FVD↓ | Sync-C↑ | Sync-D↓ | E-FID↓ | FID↓ | FVD↓ |
| AniPortrait [24] | 3.09 | 10.94 | 1.32 | 32.83 | 112.21 | 2.09 | 11.29 | 1.66 | 37.17 | 250.24 |
| VExpress [21] | 5.79 | 8.37 | 8.92 | 60.49 | 200.60 | 4.30 | 8.98 | 10.01 | 67.34 | 345.87 |
| Echomimic [4] | 5.36 | 8.99 | 1.27 | 60.82 | 240.07 | 4.16 | 9.55 | 2.87 | 63.72 | 318.08 |
| Hallo3 [3] | 6.55 | 8.49 | 1.12 | 33.98 | 153.31 | 5.57 | 8.58 | 1.51 | 40.81 | 212.91 |
| Sonic [25] | 8.35 | **6.43** | 1.22 | 29.53 | **89.34** | 6.68 | 7.31 | 1.85 | 39.89 | 224.48 |
| Fantasy Talking [11] | 3.61 | 10.78 | 1.36 | 32.64 | 103.01 | 3.14 | 10.43 | 1.77 | 37.54 | 218.43 |
| MultiTalk-single (Ours) | **8.54** | 6.69 | **1.00** | **24.01** | 95.99 | 7.07 | **7.13** | 1.41 | **32.31** | 219.19 |
| MultiTalk-multiple (Ours) | 8.53 | 6.81 | 1.24 | 27.27 | 124.06 | **7.33** | 7.18 | 1.48 | 34.08 | **184.86** |

Table 2: Quantitative comparison with other competing methods on talking body generation, including EMTD dataset.

| Methods | Sync-C↑ | Sync-D↓ | E-FID↓ | FID↓ | FVD↓ |
|---|---|---|---|---|---|
| Echomimic v2 [10] | 6.31 | 8.41 | 1.91 | 35.99 | **163.60** |
| Fantasy Talking [11] | 3.32 | 11.41 | 1.98 | 37.68 | 284.29 |
| MultiTalk-single (Ours) | 8.18 | **7.28** | 1.67 | 32.05 | 221.86 |
| MultiTalk-multiple (Ours) | **8.34** | 7.30 | **1.51** | **31.93** | 238.77 |

## 4.2 Comparisons with Competing Methods

**Quantitative Evaluation.**   To verify the effectiveness of our method, we compare it with several state-of-the-art human animation methods. For talking head generation, we compare with AniPortrait [24], VExpress [21], EchoMimic [4], Hallo3 [3] Sonic [25] and Fantasy Talking [11]. For talking body comparison, we compare with EchoMimicV2 [10] and Fantasy Talking [11].

Quantitative comparisons, including both talking head and talking body analyses, are presented in Table 1 and Table 2, respectively. Our method surpasses most other approaches across a majority of metrics, exhibiting superior performance in lip synchronization and video quality, which underscores the effectiveness of our approach.

**Qualitative Evaluation.**   To demonstrate the visual effectiveness of the proposed method, we compare and visualize the results alongside some competitive methods, as shown in Fig. 6. Upon providing instructions via a text prompt, only our method successfully responded to the instructions, highlighting its robust instruction-following capability. Additionally, our method generates fewer artifacts in the produced video, attesting to the quality of our approach.

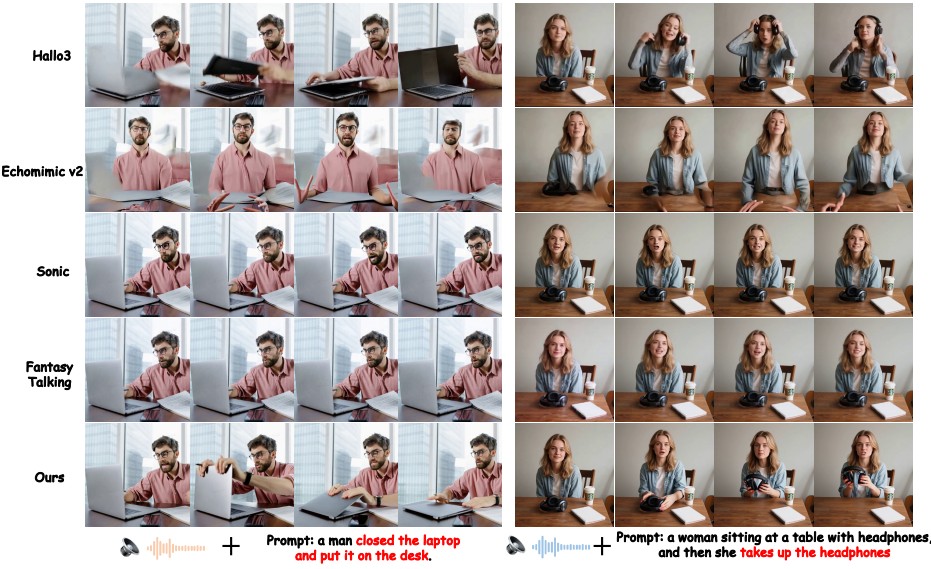

Figure 6: Qualitative comparison with other competing methods.

Table 3: Ablation study about the label range selection in L-RoPE on MTHM dataset.

| Variant | Label for video | | Label for audio | | Sync-C↑ | Sync-D↓ | E-FID↓ | FID↓ | FVD↓ |
|---|---|---|---|---|---|---|---|---|---|
| | person1 | person2 | person1 | person2 | | | | | |
| a) | 0–2 | 2–4 | 1 | 3 | 7.47 | 7.22 | 3.22 | 52.87 | 506.49 |
| b) | 0–4 | 20–24 | 2 | 22 | 7.56 | 7.13 | 3.16 | 54.20 | 508.01 |

As the first method for multi-person generation, there is no directly comparable approach available. We compare our method with the video concatenation technique, which involves generating the left and right video patches separately and subsequently concatenating them. The comparison results are presented in Fig. 7. Our method effectively handles interactive scenarios, avoiding inconsistencies between the left and right segments of the video. Besides, we also visualize the self-attention map for the specific person, highlighted in the red box. Our method can adaptively identify the localization of the person, thereby benefiting the audio binding.

## 4.3 Analyses

**Multi-stream vs Single-stream.** Our initial model for multi-stream audio training is derived from a single human animation model. To investigate whether multi-stream audio training would lead to performance degradation, we compared the performance of the single human animation model with multiple human animation models on both the talking head and talking body datasets. The results, presented in Table 1 and Table 2, show that our multiple human animation models achieve performance comparable to that of the single human animation models, indicating that multi-stream audio training does not result in model degradation.

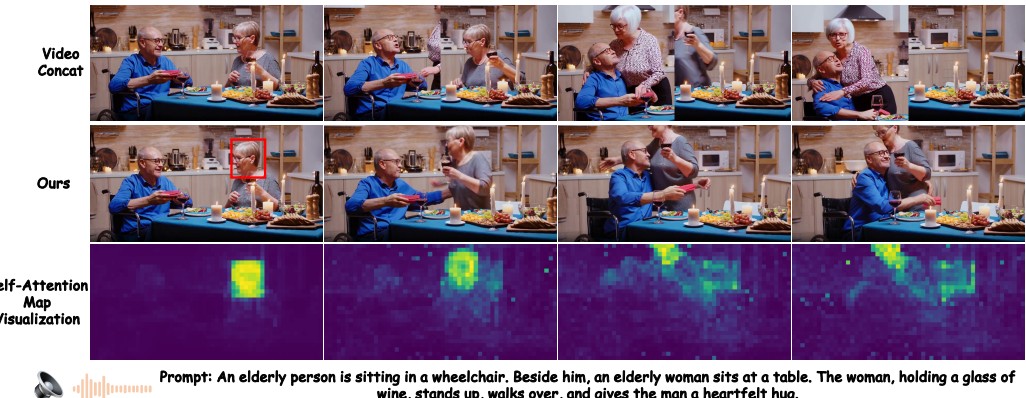

Prompt: An elderly person is sitting in a wheelchair. Beside him, an elderly woman sits at a table. The woman, holding a glass of wine, stands up, walks over, and gives the man a heartfelt hug.

Figure 7: Qualitative comparison with video concat method in multi-human animation.

**Label Selection for L-RoPE** To validate the effectiveness of L-RoPE within MultiTalk, we conduct an ablation study focusing on label range selection. The evaluation dataset is the collected conversation data, MTHM. The experimental results are presented in Table 3. These results demonstrate that different label choices for various persons yield comparable metrics, indicating that L-RoPE is not sensitive to label range variations.

Table 4: Ablation study for different audio inject strategies (Corresponding to Fig. 3).

| | Sync-C↑ | Sync-D↓ |
|---|---|---|
| a | 3.49 | 10.73 |
| b | 3.07 | 11.26 |
| c | 7.09 | 8.00 |
| d (Ours) | **7.56** | **7.13** |

**Different Audio Injection Strategies** We conducted an additional ablation study to investigate the impact of different audio injection strategies. The results are summarized in Table 4, with each row corresponding to an audio injection strategy as illustrated in Fig. 3. Strategies (a) and (b) fail to bind multi-stream audio to the corresponding video latent regions. Strategy (c) employs a hard mask-based audio binding approach, which is capable of associating multi-stream audio with different persons; however, its effectiveness is limited to videos with minimal motion. When a person exhibits extensive

movement, this strategy also results in failure cases. In contrast, our proposed L-RoPE method (d) achieves the best results across all tested scenarios, demonstrating the superiority of our approach.

Table 5: Ablation study for different training strategies.

| | Cross-attention Training | Full Parameter Training |
|---|---|---|
| MPS↑ | **59.5** | 40.5 |

**Different Training Strategies**   To quantitatively evaluate the impact of different training strategies on the model's instruction-following ability and hand/object distortion, we conducted an additional ablation study. Specifically, we utilized a reward model [48] to directly compare the cross-attention training strategy and full parameter training, using the Multi-dimensional Preference Score (MPS) as an evaluation metric. The results, shown in Table 5, demonstrate that training only the cross-attention layers leads to a higher MPS score. This provides clear quantitative evidence that optimizing only the cross-attention layers leads to better performance compared to full-parameter training, particularly when computational resources and data are limited.

## 5   Conclusion

This paper introduces a novel task: audio-driven multi-person conversational video generation, and presents a new framework, MultiTalk, to accomplish this task. Multi-stream audio conditions are effectively injected using the proposed L-PoRE method, ensuring accurate audio and person binding. Furthermore, our findings demonstrate that partial parameter training and multi-task training are essential for maintaining the instruction-following ability of the base model, equipping our model with powerful instruction-following capability.

**Limitation.** We observe that our method performs better using real audio than using synthesized audio in terms of facial expression. The reason might be that our model is trained exclusively on real audio, which typically contains rich emotional cues and natural prosody. As a result, the generated videos exhibit more expressive and realistic facial behaviors when driven by real audio. In contrast, most current TTS-generated audio lacks emotional variation and nuanced expressiveness, leading to video outputs that appear less vivid and natural We will explore ways to mitigate the gap between real and synthesized audio for animation in future work.

**Societal Impacts** This paper introduces an effective approach for audio-driven multi-person conversational video generation to the community. However, this technology also raises ethical concerns. Beyond the risk of generating fake videos of celebrities, there are broader implications, including the potential for misuse in creating deepfakes for misinformation, defamation, fraud, or harassment. Such synthetic videos could be used to impersonate individuals, manipulate public opinion, or violate privacy. These risks are not unique to our approach, but are common across the broader field of human animation and generative models.

## Acknowledgements

This work was supported in part by the National Natural Science Foundation of China (Grant No. 62372480), in part by the Guangdong Basic and Applied Basic Research Foundation (No. 2023A1515012839), in part by 2025 Tencent AI Lab Rhino-Bird Focused Research Program, and in part by HKUST-MetaX Joint Lab Fund (No. METAX24EG01-D).

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

# Appendix

## A  Task Definition

Audio-driven multi-person conversational video generation is defined as follows: Given a reference image containing multiple persons and corresponding audio streams (with a one-to-one correspondence between each person and their audio), the goal is to synthesize a video sequence in which all persons appear together in the same frame, and each person's lip movements are temporally synchronized with their respective audio input. Unlike previous single-person talking face generation tasks, this new task requires the joint modeling of multi-person interactions, spatial consistency, and audio-visual synchronization within a unified, end-to-end generative framework, requiring only a single diffusion process.

In contrast to approaches that generate videos for each person independently and subsequently composite them, this new task demands integrated modeling of multiple individuals, offering several key advantages:

- Higher computational efficiency: Only a single inference process is required, substantially reducing computational costs.
- Global consistency: The unified framework enables better control over the overall coherence of the generated content, such as coordinated camera movements, lighting, and scene dynamics.
- Enhanced interaction modeling: This approach is inherently more suitable for capturing interactions among individuals, enabling natural and contextually appropriate reactions (e.g., when one person is speaking, others can display attentive or responsive behaviors).

## B  Dataset and Implementation Details

### B.1  Dataset Details

In this paper, we utilize three distinct testing datasets: the talking head dataset, the talking body dataset, and the dual-human talking body dataset with interactive scenarios. For the talking head and talking body datasets, we employ conventional evaluation techniques for comparison with other methods. However, for the dual-human talking body dataset, where each reference image contains two persons, we evaluate Sync-C, Sync-D, and E-FID by splitting the video into two segments: the left part and the right part. Each segment contains only one person and their corresponding audio. We then average the scores of these two segments to derive the final result for this dataset. Fig.8 showcases some examples of our dual-human dataset.

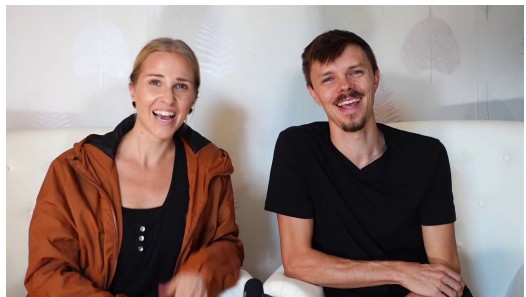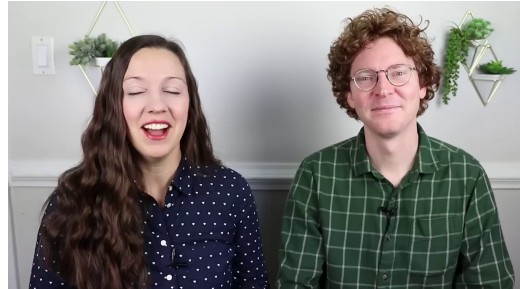

Figure 8: Some examples of our MTHM dataset.

All data used in our experiments were collected from publicly available sources on the internet. Our data collection process follows the best practices established by previous works [49, 50, 51], ensuring that our methods are consistent with the standards in the community. All data sources are under the CC BY 4.0 International license. Our dataset comprises approximately 2,700 unique subjects, with approximately 71% male and 29% female. The distributions of age and race are presented in Table 6 and 7, respectively.

Table 6: The distributions of the age of the training dataset.

| Age | 0-9 | 10-19 | 20-29 | 30-39 | 40-49 | 50-59 | 60-69 | 70+ |
|---|---|---|---|---|---|---|---|---|
| Percentage | 0% | 0.02% | 20.77% | 57.11% | 19.03% | 2.95% | 0.12% | 0% |

Table 7: The distributions of race of the training dataset.

| Race | white | black | middle eastern | asian | latino hispanic | indian |
|---|---|---|---|---|---|---|
| Percentage | 61.55% | 6.22% | 4.89% | 21.76% | 4.77% | 0.81% |

For data preprocessing, we closely follow the procedures described in [49] and further filter out samples exhibiting large facial movements or unsynchronized speech and mouth motion [52]. This ensures the high quality and reliability of our dataset.

## B.2 Sample Details

In all the experiments and evaluations conducted within this paper, we utilize 40 sampling steps. To filter out undesired variations in diffusion models, we employ the following negative prompt during sampling: "bright tones, overexposed, static, blurred details, subtitles, style, works, paintings, images, static, overall gray, worst quality, low quality, JPEG compression residue, ugly, incomplete, extra fingers, poorly drawn hands, poorly drawn faces, deformed, disfigured, misshapen limbs, fused fingers, still picture, messy background, three legs, many people in the background, walking backwards." Additionally, we employ Qwen-VL for reference image captioning.

## B.3 Inference Time

Although our method introduces an additional audio condition, the computational time required to pass through the DiT backbone remains the same as in Wan2.1, and it requires 40 steps for inference. Furthermore, all acceleration strategies available for Wan2.1—such as TeaCache and model distillation—are also applicable to our approach. For example, when employing a distilled model (such as lightx2v), the total number of inference steps is reduced to 4 steps per video.

## C Analyses

### C.1 Full Parameter Training vs Cross-attention Training

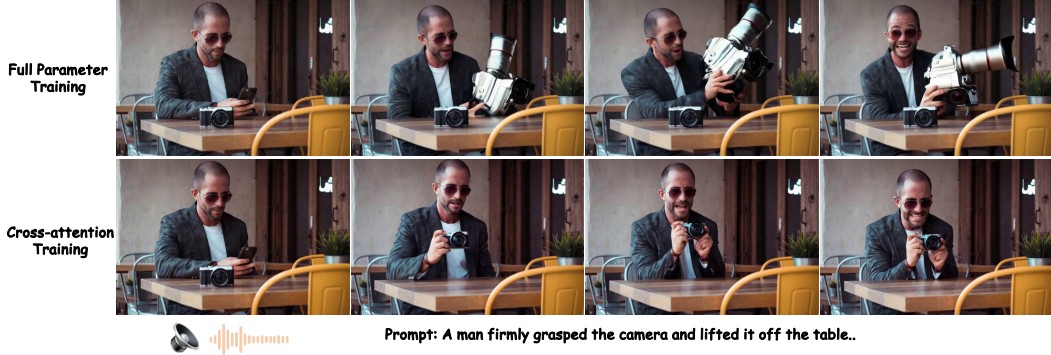

Figure 9: Comparison between full parameter training and cross-attention training.

We compare full parameter training with fine-tuning only the audio cross-attention layer. Our findings indicate that network training parameters are crucial. When compute resources and data are limited, fully parameterized training can lead not only to degradation in the model's instruction-following ability, especially for motion and interaction, but also to hand and object distortion. Conversely, training only the audio cross-attention does not result in these issues, and the instruction-following ability of the base model is well preserved. The comparison results between full parameter training

and cross-attention training are shown in Fig. 9. It can be seen that full parameter training degrades the model's instruction-following ability and causes hand distortion.

## C.2 Long Video Generation

Utilizing the autoregressive-based method facilitates the long video generation of our method. The experimental results for long video generation are shown in Fig.10. This example shows a generated result containing 305 frames.

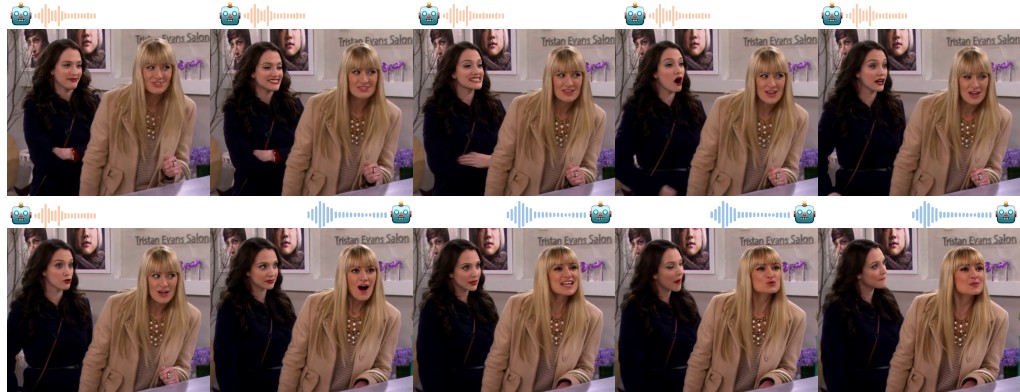

Figure 10: The generation result of long videos.

## C.3 Emotional Expressions

We use the same reference image and specify emotional (e.g., angry, sad, happy) via the text prompt. Our model successfully generates videos with the corresponding emotional expressions, as shown in Fig. 11.

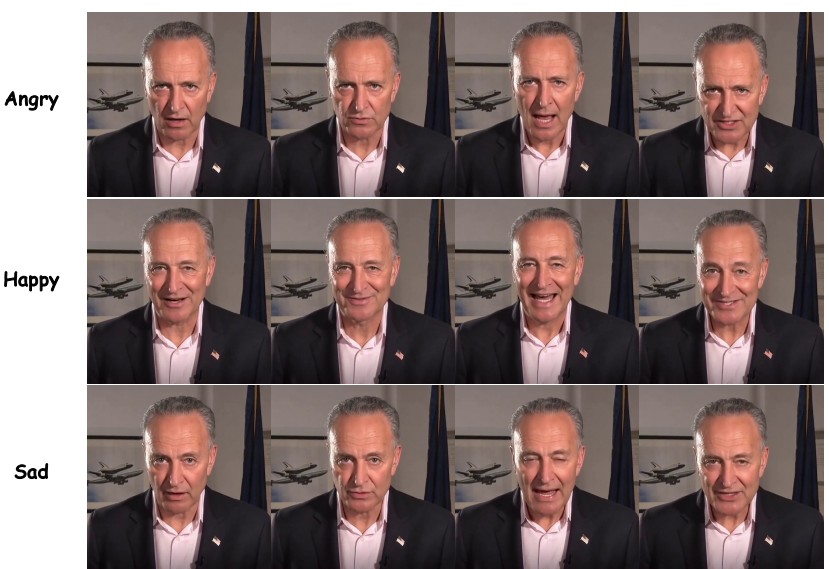

Figure 11: Generation results for videos with different emotions using the same reference image.

## C.4 Generalization to More Speakers

In the two-speaker setting, we assign distinct, non-overlapping ranges of video and audio labels to each individual (e.g., video labels 0–4 for person 1 and 20–24 for person 2; audio labels 2 for

person 1 and 22 for person 2). For scenarios involving more individuals, we conducted additional experiments to verify that this labeling scheme can be extended by assigning new, non-overlapping ranges (e.g., video labels 40–44 and audio label 42 for a third person). This flexible approach enables the model to accommodate a greater number of speakers simply by expanding the label ranges for both video and audio streams, which demonstrates the generalizability of our L-RoPE framework. The visualization results for scenarios involving three speakers are shown in Fig. 12. Importantly, the L-RoPE extension strategy remains effective even when the number of persons during inference differs from that during training. By assigning dedicated label ranges, each person's lip movements are accurately aligned with their respective audio streams.

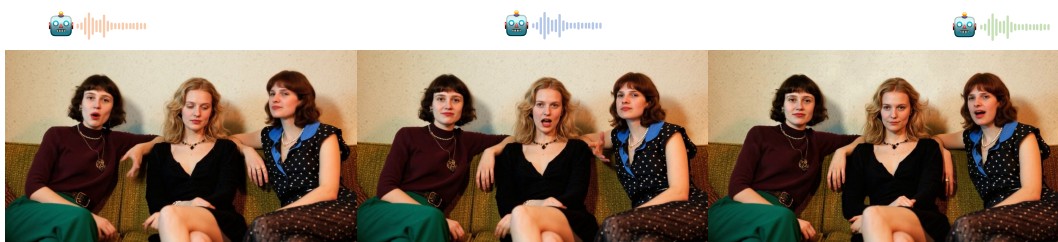

Figure 12: The generation results generalize to scenarios with three persons.

