# OpenReview forum: "Let Them Talk: Audio-Driven Multi-Person Conversational Video Generation"
_NeurIPS.cc/2025/Conference — NeurIPS 2025 poster_

### Official Review · Reviewer_s1Za · 2025-06-08

**Clarity:** 2
**Significance:** 1
**Originality:** 2
**Rating:** 3
**Confidence:** 5

**Summary:**

This paper approaches the task of multi-person audio-to-human generation using video Diffusion Transfomers. It specifically aims to address the issues of having multiple people present in the output video, which previous methods do not handle well. This work proposes the use of Label Rotary Position Embedding (L-RoPE) which helps to bind audio streams to the correct tokens (and therefore regions) in the diffusion process.

**Questions:**

To be accepted, the following questions on data must be answered:

- How is the data obtained?
- Where is it sourced from?
- Do all of these sources have a permissive license?
- How is consent obtained from each of the subjects?
- How many subjects are there?
- What is the demographic distribution of the subjects (e.g. age, race, gender etc)?

Furthermore, I have the following questions:

- Do the authors want to claim improved performance in the single speaker setting?
- If so, what are the differences beyond L-RoPE, how much does the newly collected dataset contribute to the performance, rather than methodological differences?
- If not, and only multi-speaker is claimed, how well do the other methods perform when trained with the multi-person data collected in this method, and how does the proposed method perform without L-RoPE?

If the work can be shown to be better in either setting through fair evaluation, and the data concerns are addressed, the paper could be accepted.

**Ethical Concerns:**

["Major Concern: Data privacy, copyright, and consent", "Major Concern: Data quality and representativeness"]

**Final Justification:**

Thanks to the authors for their clarification. I no longer have any issues with the technical content of the paper, provided the extra experiments are included. However, I still have ethical concerns around the data, as the collection does violate YouTube's ToS. I will allow the ethics reviewers to make the final decision here.

**Limitations:**

More discussion is needed, see weaknesses and questions.

**Paper Formatting Concerns:**

No formatting concerns.

**Quality:**

2

**Strengths And Weaknesses:**

**Strengths**

The paper is well motivated, it is true that there are difficulties around multi-speaker audio-to-human video generation. This is justified in the paper, and a novel component L-RoPE is proposed to address the issue. L-RoPE is described well and seems to make sense theoretically. The results shown, both quantitatively and qualitatively are very impressive.

**Weaknesses**

There are major concerns with this paper related to the data. There is a severe lack of discussion on this. It is stated in the paper:

"We collect a video dataset of about 2K hours for the first stage training, which covers the
face or body of a single talking person. We also collect about 200K video clips that contain multiple
events and human-object/environment interactions. The average clip duration is about 10 seconds.
For the second stage training, we collect 100 hours of videos consisting of conversations between two
persons."

This is problematic on several fronts. From a reproducability perspective, there is not enough information to reproduce this dataset. Furthermore, the following are in the Neurips ethics guidelines:

- Privacy: Datasets should minimize the exposure of any personally identifiable information, unless informed consent from those individuals is provided to do so.
- Consent: Any paper that chooses to create a dataset with real data of real people should ask for the explicit consent of participants, or explain why they were unable to do so.
- Copyright and Fair Use: While the norms of fair use and copyright in machine learning research are still evolving, authors must respect the terms of datasets that have defined licenses (e.g. CC 4.0, MIT, etc).
- Representative evaluation practice:  When collecting new datasets or making decisions about which datasets to use, authors should assess and communicate the degree to which their datasets are representative of their intended population. Claims of diverse or universal representation should be substantiated by concrete evidence or examples.

Without further information, all of these are significant concerns.

In terms of evaluations. In the single speaker setting, it is not clear how this work will differ from previous DiT works such as FantasyTalking and Hallo3. It seems to me that the only difference is the L-RoPE module, which is designed for the multi-speaker setting. If the authors want to show that it improves the single speaker case, an ablation study is needed. This is because the data used is different from the baselines, so it is not a fair comparison. If this is not being claimed then Tables 1 & 2 should be removed, and in its place the exitsing models should be shown **not** to work in the multi-speaker case, when trained on multi-speaker data as the proposed method has been. In either case, the evidence is lacking in this work.

It is not clear what Table 3 is trying to show. It is claimed that it is "To validate the effectiveness of L-RoPE within MultiTalk" however, it does not seem to do this. An ablation to show this should be with L-RoPE vs without. The table appears to show the effects of the range parameter, which while interesting, does not support what is claimed.

Overall, I am not convinced that the main improvements in this paper are not just due to better dataset collection.

---

> ### Author Rebuttal · Authors · 2025-07-31
>
> We appreciate the reviewers’ positive feedback, including recognition of our well-motivated methodology, the novelty of L-RoPE, and our impressive quantitative and qualitative results. Below, we address the specific concerns raised by the reviewers individually.
>
> ### Q1: More details of our dataset.
>
> Thank you for raising these important questions regarding our dataset. We appreciate the opportunity to clarify the data-related aspects of our work.
> All data used in our experiments were collected from publicly available sources on the internet. We only downloaded data from platforms that explicitly permit data download for research purposes, and we strictly avoided any sources that prohibit such use. Our data collection process follows the best practices established by previous works [1]-[3], ensuring that our methods are consistent with the standards in the community.
> All data sources are under the CC BY 4.0 International license that allows for academic research and non-commercial use. We comply with the platform’s terms of service and applicable privacy regulations during the data collection and usage process.
> Our dataset comprises approximately 2,700 unique subjects, with approximately 71\% male and 29\% female. The distributions of age and race are presented in the following tables.
> We have made efforts to ensure diversity in age, race, and gender to the best extent possible based on the available metadata and annotations. We acknowledge that perfect demographic balance is challenging due to the nature of publicly available data, but we have taken steps to maximize representativeness.
> We strictly followed the NeurIPS ethics guidelines regarding privacy, consent, and copyright. All data is used strictly for research purposes and will not be commercialized.
>
> | Age        | 0-9 | 10-19 | 20-29  | 30-39  | 40-49  | 50-59 | 60-69 | 70+ |
> |------------|-----|-------|--------|--------|--------|-------|-------|-----|
> | Percentage | 0.00%  | 0.02% | 20.77% | 57.11% | 19.03% | 2.95% | 0.12% | 0.00%  |
>
>
> |    Race    |  white | black | middle eastern |  asian | latino hispanic | indian |
> |:----------:|:------:|:-----:|:--------------:|:------:|:---------------:|:------:|
> | Percentage | 61.55% | 6.22% |      4.89%     | 21.76% |      4.77%      |  0.81% |
>
> For data preprocessing, we closely follow the procedures described in [1] and further filter out samples exhibiting large facial movements or unsynchronized speech and mouth motion [4]. This ensures the high quality and reliability of our dataset.
>
> We acknowledge the importance of reproducibility and are committed to supporting the community.
> We will release the MTHM test dataset, along with the corresponding code and model weights, to the research community to encourage reproducibility and facilitate further research and development in audio-driven human animation.
>
>
>
> [1] Koala-36M: A Large-scale Video Dataset Improving Consistency between Fine-grained Conditions and Video Content
>
> [2] Openvid-1m: A large-scale high-quality dataset for text-to-video generation
>
> [3] MiraData: A Large-Scale Video Dataset with Long Durations and Structured Captions
>
> [4] Out of time: automated lip sync in the wild
>
>
>
> ### Q2: Clarification of our evaluation.
>
> There appears to be a misunderstanding regarding our paper. The "MultiTalk single" entries in Tables 1 and 2 report results from our pre-trained single-person model on single-person datasets, without using L-RoPE. Similarly, "MultiTalk multiple" refers to our fine-tuned model, also evaluated without L-RoPE. In both cases, L-RoPE is not involved in single-person inference; it is only required for multi-person generation. We present both "MultiTalk single" and "MultiTalk multiple" in the table to demonstrate that L-RoPE does not degrade single-person performance, and our strong results in single-person scenarios are not due to L-RoPE.
>
> Regarding differences between "MultiTalk single" and the FantasyTalking and Hallo3:
>
> (1). Model architecture: Our method uses Wan2.1 as the video backbone, while Hallo3 uses CogVideo. Although both are DiT-based, there is a significant performance gap between different foundation models. Hallo3 also incorporates an identity preservation module, which we do not. Compared to FantasyTalking, we employ a different audio encoding method, and FantasyTalking introduces a Face-Cross-Attention mechanism, which our model does not have.
>
> (2). Training strategy: We use partial parameter training and a multi-task training strategy, which are different from those in Hallo3 and FantasyTalking.
>
> On the issue of data fairness, it is common practice in this field to compare methods using their official open-source models, as each work is trained on its own collected dataset. For example, Hallo3 uses 1,200 hours of YouTube data + 2,346 hours of film video, FantasyTalking leverages large-scale internet data, OmniHuman-1 uses 18.7K hours of data, and Hunyuan-Avatar uses 1,250 hours of collected data. Given these differences, we compare directly with the available open-source checkpoints, following standard practice.
>
> ### Q3: Explanation of Table 3.
>
> In our L-RoPE design, we assign distinct video and audio label ranges to different persons (e.g., video labels 0–4 for person 1 and 20–24 for person 2; audio labels 2 for person 1 and 22 for person 2). Table 3 investigates the impact of using different label ranges for each person. The results indicate that varying the label assignments yields comparable performance results, which demonstrates that L-RoPE is robust to the choice of label ranges and not sensitive to this hyperparameter.
>
> We acknowledge that Table 3 does not provide a direct comparison between models with and without L-RoPE. However, such an ablation is not feasible in the multi-person setting, as it is not possible to generate audio and lip synchronized videos for two individuals without the L-RoPE mechanism.

---

> > ### Comment · Reviewer_s1Za · 2025-08-03
> >
> > Many thanks to the authors for this rebuttal, particularly for including more details about the dataset. Thank you also for the clarification around the tables, I am now satisfied with these. However, I still have several concerns.
> >
> > Data
> >
> > - **Representation:** The dataset does contain significant biases, particularly regarding the gender split. However, if the ethics reviewers are okay with this, then I would not consider it a barrier to acceptance.
> > - **Sources:** Thank you for clarifying the licenses used for this data. You say you obtained these from platforms with permissive research licenses. Please specify which platforms you used. As mentioned by the ethics reviewer, please release the details of the sources and licenses. A list of platforms used (e.g., YouTube) and the license terms would be sufficient here.
> >
> > Model Differences
> > - **Architectures:** With the difference between MultiTalk Single and Hallo3, I would not consider using a different backbone as a research contribution. Stating that you do not use their components, which they show as an improvement, also is not a contribution.
> > - **L-RoPE:** This leaves L-RoPE as the only novel component of this work. As this is not used in the single-speaker setting, the authors cannot make any claims here. Essentially, what is shown here is that Wan2.1>CogVideo, which is a swap in backbone, and is not a contribution to the field.
> >
> >
> > Evaluation
> > - **Open-Source Checkpoints:** I agree that it is common practice to compare with models using the publicly available checkpoints. However, this only applies to models performing the same task (e.g. Single Speaker). To claim improvements in the multispeaker case, you would need to train these models on multi-speaker data, or at least fine-tune. It is a different task, so it is not a fair comparison. How can the other models be expected to perform in the multi-speaker case if they have never seen any examples? As you cannot claim contributions in the single-speaker case, as mentioned above. This experiment would be absolutely critical.
> >
> > Provided that the authors include a list of data sources and licenses, I will withdraw any ethics concerns. However, for the above reasons, there is not any novel contribution in the single speaker case, and the contribution for the multi-speaker case is not properly evaluated, I would still not recommend accepting this work at this time.

---

> > > ### Author Response · Authors · 2025-08-04
> > >
> > > ## Clarification of Data
> > >
> > > Thank you for your comment regarding the dataset.
> > > We acknowledge that there is an imbalance in the gender distribution within our dataset. However, the primary focus of our work is on audio-driven video generation, specifically the consistency between audio and lip movements, rather than gender-specific characteristics. The model is designed to learn generalizable audio-visual synchronization patterns that are not inherently dependent on gender.
> > >
> > > For the data source, all data used in our study were collected from publicly available videos on YouTube, in accordance with YouTube’s Terms of Service. The videos used are those that are publicly accessible and are intended for research purposes only.
> > >
> > >
> > >
> > > ## Clarification of Contributions and Novelty
> > >
> > > Thank you for your comments regarding the novelty of our contributions. We would like to clarify that our work does not claim innovation in single-person animation or in simply changing the backbone architecture. The main contributions of our work are as follows:
> > >
> > > **Novel Task Definition**:
> > > We introduce the new task of audio-driven multi-person conversational video generation, which goes beyond the scope of existing single-person talking face or animation models.
> > >
> > > **L-RoPE for Multi-Person Audio Binding**:
> > > To address the core challenge of accurate audio-person binding in multi-person scenarios, we propose the Label Rotary Position Embedding (L-RoPE) method. This component is specifically designed for the multi-person setting and is a key technical contribution of our work.
> > >
> > > **Training Strategies for Instruction-Following**:
> > > We systematically investigate different training strategies and find that partial parameter training and multi-task training are crucial for preserving the instruction-following ability of the base model in the multi-person context.
> > >
> > >
> > > ## Clarification of Evaluation
> > >
> > > Thank you for your comments regarding the proper evaluation of the multi-speaker case.
> > > However, we believe that training or fine-tuning existing single-person animation models on multi-speaker data is neither feasible nor appropriate for several reasons.
> > > 1) The proposed L-RoPE is specifically designed for the multi-speaker task. Training or fine-tuning existing single-person animation models on multi-speaker data with L-RoPE would only enable comparisons among different single-speaker models, but would not provide a meaningful evaluation in the multi-speaker setting.
> > > 2) Current single-speaker models—such as Hallo3 and FantasyTalking—are specifically designed for single-person animation. Their architectures often include identity preservation modules and other components that are not readily extensible to multi-speaker scenarios. Adapting these models would require substantial architectural redesign, particularly to address challenges such as identity and audio-related issues in multi-person settings. This is a non-trivial task that goes far beyond simple fine-tuning, even after equipping our proposed L-RoPE. Instead, it involves the design of entirely new methods specifically for the proposed multi-speaker task, which would constitute a separate research paper.
> > > 3) Most open-source single-speaker methods only provide inference code, without access to the full training pipeline or critical implementation details. Consequently, it is not feasible to first redesign these non-extensible modules for multi-speaker scenarios and then retrain or fine-tune these models on multi-speaker data within the current discussion period.
> > >
> > > Multi-person conversational video generation is a novel task introduced in our work. To the best of our knowledge, there are no prior works specifically designed for this task. The absence of prior multi-person generation methods further demonstrates the novelty and contribution of our approach.

---

> > > > ### Comment · Reviewer_s1Za · 2025-08-04
> > > >
> > > > Thank you for your reply. After reading this, I am more inclined to accept the novelty of the work. However, I am still not convinced by the evaluation.
> > > >
> > > > **L-RoPE:** You state, "an ablation is not feasible in the multi-person setting, as it is not possible to generate audio and lip synchronised videos for two individuals without the L-RoPE mechanism." I do not see why this is the case. It should be possible to include both audio streams for audio cross-attention using any number of mechanisms (concatenation, averaging, or a small MLP) rather than the more sophisticated version. As L-RoPE seems to be the main contribution of this work, I would expect these naive baselines as a minimum.
> > > >
> > > > **Training Strategies for Instruction-Following:** The authors claim these have been systematically investigated; however, there is not enough evidence for this. There is no quantitative ablation for these components, and only a single image of qualitative evidence is in the supplementary material. This could well have been cherry-picked without more solid evidence. To make claims on these strategies, it would be essential to include quantitative ablations.
> > > >
> > > > I am also seriously concerned about the source of the data. YouTubes terms of service **explicitly prohibit** data collection in this way, with a particular emphasis on disallowing the use of faces. See https://www.youtube.com/static?template=terms&hl=en&gl=SE (relevant points summarised below):
> > > >
> > > > **Permissions and Restrictions:**
> > > >
> > > > The following restrictions apply to your use of the Service. You are not allowed to:
> > > >
> > > > 1) access, reproduce, download, distribute, transmit, broadcast, display, sell, license, alter, modify or otherwise use any part of the Service or any Content
> > > > 3) access the Service using any automated means (such as robots, botnets or scrapers) except: (a) in the case of public search engines, in accordance with YouTube’s robots.txt file; (b) with YouTube’s prior written permission; or (c) as permitted by applicable law;
> > > > 4) collect or use any information that might identify a person (for example, harvesting usernames or faces), unless permitted by that person or allowed under section 3 above;
> > > >
> > > > I am also concerned by the statement from the authors that: "the primary focus of our work is on audio-driven video generation, specifically the consistency between audio and lip movements, rather than gender-specific characteristics". This is not the point of representation in datasets. Collecting a fair and representative dataset is essential to prevent biases in the model. I understand that it is hard to guarantee fair representation in large datasets, so I did not raise an objection to this; however, the above statement leads me to believe that due care may not have been taken.
> > > >
> > > > I would like to re-raise the points with the ethics reviewers.

---

> > > > > ### Author Response · Authors · 2025-08-05
> > > > > **Response to Reviewer s1Za (Part 1/2)**
> > > > >
> > > > > We appreciate the reviewer’s recognition of the novelty of our work, as reflected in the statement, “I am more inclined to accept the novelty of the work.”
> > > > >
> > > > >
> > > > > ### 1.Clarification of Our Previous Response
> > > > >
> > > > > We would like to clarify a misunderstanding regarding our previous response. At no point did we state that ablation studies in the multi-person setting are impossible. Our original statement—“We acknowledge that Table 3 does not provide a direct comparison between models with and without L-RoPE. However, such an ablation is not feasible in the multi-person setting, as it is not possible to generate audio and lip synchronized videos for two individuals without the L-RoPE mechanism.”—was specifically intended to address your suggestion that “An ablation to show this should be with L-RoPE vs without.”
> > > > >
> > > > > Our intent was to convey that a direct comparison of “with L-RoPE vs. without L-RoPE” is infeasible in the multi-person scenario because, without L-RoPE, it is not possible to generate videos with correct audio-visual synchronization for multiple individuals. This does not mean that ablation studies in general cannot be performed, but rather that this specific comparison—“with L-RoPE vs. without L-RoPE”—is not feasible in the context of multi-person generation.
> > > > >
> > > > >
> > > > > ### 2.Clarification on YouTube Terms of Service and Data Usage
> > > > >
> > > > > Thank you for sharing the relevant YouTube Terms of Service URL and raising your concerns.
> > > > > We would like to respectfully clarify a misrepresentation regarding YouTube’s Terms of Service as referenced by the reviewer. The summary provided omits critical clauses that are central to the interpretation of YouTube’s permissions and restrictions.
> > > > >
> > > > > Specifically, the full text of the Terms of Service states:
> > > > >
> > > > > 1. access, reproduce, download, distribute, transmit, broadcast, display, sell, license, alter, modify or otherwise use any part of the Service or any Content **except: (a) as expressly authorized by the Service; or (b) with prior written permission from YouTube and, if applicable, the respective rights holders;**
> > > > >
> > > > > 2. circumvent, disable, fraudulently engage with, or otherwise interfere with any part of the Service (or attempt to do any of these things), including security-related features or features that (a) prevent or restrict the copying or other use of Content or (b) limit the use of the Service or Content;
> > > > >
> > > > > 3. access the Service using any automated means (such as robots, botnets or scrapers) except (a) in the case of public search engines, in accordance with YouTube’s robots.txt file; or (b) with YouTube’s prior written permission;
> > > > > ……
> > > > >
> > > > > We comply with the platform’s terms of service and applicable privacy regulations during the data collection and usage process. We only downloaded data from platforms that explicitly permit data download for research purposes, and we strictly avoided any sources that prohibit such use. Our data collection process follows the best practices established by previous works [1]-[3], ensuring that our methods are consistent with the standards in the community.
> > > > > All data sources are under the CC BY 4.0 International license that allows for academic research and non-commercial use.
> > > > > We strictly followed the NeurIPS ethics guidelines regarding privacy, consent, and copyright. All data is used strictly for research purposes and will not be commercialized.
> > > > >
> > > > > [1] Koala-36M: A Large-scale Video Dataset Improving Consistency between Fine-grained Conditions and Video Content
> > > > >
> > > > > [2] Openvid-1m: A large-scale high-quality dataset for text-to-video generation
> > > > >
> > > > > [3] MiraData: A Large-Scale Video Dataset with Long Durations and Structured Captions

---

> > > > > > ### Author Response · Authors · 2025-08-05
> > > > > > **Response to Reviewer s1Za (Part 2/2)**
> > > > > >
> > > > > > ### 3.Ablation Study for Audio Injection
> > > > > >
> > > > > > We conducted an additional ablation study to investigate the impact of different audio injection strategies. The results are summarized in the table below, with each row corresponding to an audio injection strategy as illustrated in Figure 3. Strategies (a) and (b) fail to bind multi-stream audio to the corresponding video latent regions. Strategy (c) employs a hard mask-based audio binding approach, which is capable of associating multi-stream audio with different persons; however, its effectiveness is limited to videos with minimal motion. When a person exhibits extensive movement, this strategy also results in failure cases. In contrast, our proposed L-RoPE method (d) achieves the best results across all tested scenarios, demonstrating the superiority of our approach. We will provide these results in the supplementary materials of the updated version.
> > > > > >
> > > > > > |          | Sync-C↑ | Sync-D↓ |
> > > > > > |:--------:|:-------:|:-------:|
> > > > > > |     a    |   3.49  |  10.73  |
> > > > > > |     b    |   3.07  |  11.26  |
> > > > > > |     c    |   7.09  |   8.00  |
> > > > > > | d (Ours) |   7.56  |   7.13  |
> > > > > >
> > > > > > Additionally, we provide a comparison between L-RoPE and a video concatenation approach utilizing a single-person animation method in Figure 7 of the main paper.
> > > > > >
> > > > > > ### 4.Ablations Study for Training Strategies
> > > > > >
> > > > > >
> > > > > > Thank you for your valuable feedback regarding the need for quantitative evidence to support our claims about training strategies.
> > > > > >
> > > > > > In response to your suggestion, we have conducted an additional ablation study to quantitatively evaluate the impact of different training strategies on the model’s instruction-following ability and on hand/object distortion. Specifically, we utilized a reward model [1] to directly compare the cross-attention training strategy and full parameter training, using the Multi-dimensional Preference Score (MPS) as an evaluation metric.
> > > > > > The results, shown in the following table, demonstrate that training only the cross-attention layers leads to a higher MPS score. This provides clear quantitative evidence that optimizing only the cross-attention layers leads to better performance compared to full-parameter training, particularly when computational resources and data are limited.
> > > > > >
> > > > > > We will include these quantitative ablation results in the revised manuscript to provide a more solid foundation for our claims.
> > > > > >
> > > > > > |      | Cross-attention Training | Full Parameter Training |
> > > > > > |:----:|:------------------------:|:-----------------------:|
> > > > > > | MPS↑ |         **59.5**         |           40.5          |
> > > > > >
> > > > > > [1] Learning Multi-dimensional Human Preference for Text-to-Image Generation.

---

> > > > > > ### Comment · Reviewer_s1Za · 2025-08-05
> > > > > >
> > > > > > Thank you for the further clarification
> > > > > >
> > > > > >
> > > > > > ## L-RoPE
> > > > > >
> > > > > > You state that "it is not possible to generate audio and lip-synchronised videos for two individuals without the L-RoPE mechanism", perhaps I am misunderstanding here, but I do not see why this is the case. I could agree that the results may not look good or work well. That's expected, and it would need to be shown to validate that the L-RoPE works. However, there should be nothing to stop you from training a model without this mechanism. I would appreciate clarification here. Is there some architectural reason that removing the L-RoPE mechanism would not allow video generation, or is it that the results would be expected not to accurately sync both subjects?
> > > > > >
> > > > > > ## YouTube ToS
> > > > > >
> > > > > > Respectfully, this is also a misrepresentation of the terms of service. Clause 1 actually says
> > > > > >
> > > > > > 1. access, reproduce, download, distribute, transmit, broadcast, display, sell, license, alter, modify or otherwise use any part of the Service or any Content except: (a) as expressly authorized by the Service; $\cancel{\text{or}}$ (b) with prior written permission from YouTube and, if applicable, the respective rights holders;
> > > > > >
> > > > > > Without the or in between a and b, this means that anyone wishing to use YouTube data must have explicit written permission from YouTube. I had omitted this as it was assumed the authors would mention this if they had it. If the authors have obtained this written permission, could they please provide copies?
> > > > > >
> > > > > > Clause 3 also prohibits scrapers, and it is very hard to believe that the authors were able to collect 200,000 videos without doing this. Clause 4 prohibits the collection of faces explicitly without written permission.
> > > > > >
> > > > > > I appreciate that using YouTube for data has been a common practice for many datasets in the past. However, for the past few years, NeurIPS has decided to hold itself to higher standards around ethics and data legality, and so collection in this way would no longer be acceptable.
> > > > > >
> > > > > > ## Ablations
> > > > > >
> > > > > > Thank you for including these they certainly help strengthen your case. Please can you confirm that these were performed on the full MTHM dataset.

---

> > > > > > > ### Comment · Area_Chair_Mi7E · 2025-08-05
> > > > > > > **A couple clarifications**
> > > > > > >
> > > > > > > Dear authors and reviewers,
> > > > > > >
> > > > > > > I would like to respectfully point out that the YT Terms of Service (https://www.youtube.com/static?template=terms) does indeed include the **"or"** in Clause 1 (Reviewer s1Za seems to incorrectly claim that the **"or"** is not there):
> > > > > > >
> > > > > > > 1. access, reproduce, download, distribute, transmit, broadcast, display, sell, license, alter, modify or otherwise use any part of the Service or any Content except: (a) as expressly authorized by the Service; **or** (b) with prior written permission from YouTube and, if applicable, the respective rights holders;
> > > > > > >
> > > > > > > The authors claim that *"..the ethics reviewer having acknowledged that our data collection and use do not present ethical issues"*. However, I don't think this is the case; from what I read from the ethics review comments, the ethics reviewer stated specific concerns, and the authors responded, but I don't think the ethics reviewer has cleared the data collection of ethical issues yet. Note that the ethics reviewer cannot see comments outside of the ethics review comment thread; please ensure the ethics reviewer has all the information they need.
> > > > > > >
> > > > > > > In any case, let's restrict discussion of ethical issues with the data collection to the thread with the ethics reviewer, and please focus discussion on other aspects of the paper that may require additional discussion between authors and reviewers. For example, I think there may be some issues to clarify regarding Reviewer s1Za's question about ablation without L-RoPE.
> > > > > > >
> > > > > > > Thanks for the robust discussion, and for your efforts and consideration,
> > > > > > > AC

---

> > > > > > > > ### Comment · Reviewer_s1Za · 2025-08-05
> > > > > > > >
> > > > > > > > Thank you for weighing in here. However, I am very confused. The word **or** is certainly not in there. It says (in full):
> > > > > > > >
> > > > > > > > Permissions and Restrictions
> > > > > > > >
> > > > > > > > You may access and use the Service as made available to you, as long as you comply with this Agreement and the law. You may view or listen to Content for your personal, non-commercial use. You may also show YouTube videos through the embeddable YouTube player.
> > > > > > > >
> > > > > > > > The following restrictions apply to your use of the Service. You are not allowed to:
> > > > > > > >
> > > > > > > > access, reproduce, download, distribute, transmit, broadcast, display, sell, license, alter, modify or otherwise use any part of the Service or any Content except: (a) as specifically permitted by the Service;  (b) with prior written permission from YouTube and, if applicable, the respective rights holders; or (c) as permitted by applicable law;
> > > > > > > >
> > > > > > > > Nonetheless, this is not necessary as clauses 3 and 4 are also violated here.

---

> ### Comment · Reviewer_s1Za · 2025-08-05
>
> Apologies, I have just used a VPN and found out this is location-dependent. I will discuss with the ethics reviewers in the other thread

---

> ### Author Response · Authors · 2025-08-06
>
> ### L-RoPE
>
> Thank you for your question regarding the necessity of the L-RoPE mechanism for generating audio and lip-synchronized videos for multiple individuals.
>
> After removing L-RoPE, the model is unable to achieve accurate synchronization between each audio stream and the corresponding person’s lip movements. The reason is that, without L-RoPE, there is no effective mechanism to bind each audio stream to the correct visual region in the attention map. As a result, the attention weights for each audio stream are distributed more uniformly across the entire video, which prevents the model from learning distinct audio-person associations. This case corresponds to Figure 3(a).
>
>
> Beyond simply removing L-RoPE, we also experimented with alternative approaches, as shown in Figure 3(b) and (c):
>
> **Strategy (b)**: This approach also fails to establish effective audio-person binding, resulting in poor synchronization.
>
> **Strategy (c)**: This mask-based method can achieve audio-person association in simple cases with minimal motion, but it fails when there is significant movement or complex interactions.
>
> We present the quantitative results for each of these approaches on the full MTHM dataset in the following table, which further demonstrates the effectiveness of L-RoPE for robust multi-person video generation. We will include these clarifications and results in the revised manuscript.
>
> |          | Sync-C↑ | Sync-D↓ |
> |:--------:|:-------:|:-------:|
> |     a    |   3.49  |  10.73  |
> |     b    |   3.07  |  11.26  |
> |     c    |   7.09  |   8.00  |
> | d (Ours) |   7.56  |   7.13  |
>
> ### Ablations Study for Training Strategies
>
> Yes, we confirm that the provided ablation studies for training strategies were performed on the full MTHM dataset. Thank you for your attention to this detail.

---

> > ### Comment · Reviewer_s1Za · 2025-08-06
> >
> > Thank you again. This helps clarify the contribution of L-RoPE, it would be beneficial to state 3a) is the version of the model without L-RoPE explicitly in the Table.
> >
> > I no longer have any objection to accepting this paper based on evaluation or contribution, but will await the ethics desision before changing my ratings, as I still have serious concerns here.

---

### Official Review · Reviewer_34Jy · 2025-06-28

**Clarity:** 3
**Significance:** 2
**Originality:** 2
**Rating:** 3
**Confidence:** 5

**Summary:**

The paper seeks to use multi-stream audio to animate multi-person from an image. It proposes combining Adaptive Person Localization and L-RoPE to achieve this goal. The model is trained on 100 hours of videos and achieves better results on benchmarks like HDTF, CelebV-HQ, and EMTD than the baselines.

**Questions:**

Since all cases have a maximum of two person, I wonder how many people this method can scale to at most, and can it handle the situation where the masks of multiple people overlap?

**Ethical Concerns:**

["NO or VERY MINOR ethics concerns only"]

**Final Justification:**

Thanks for the author's reply, which has addressed some of my concerns. Finally, I suggest that the author provide more examples of people moving and scenes with more than two persons, as these are sufficient to prove what the paper claims. Currently, due to the lack of these examples, I maintain my rating (3 Borderline Reject), but I am not opposed to accepting this paper.

**Limitations:**

yes

**Quality:**

3

**Strengths And Weaknesses:**

**Strengths**
* This task of driving multiple person has moved forward from the current audio-driven task.
* The L-RoPE could differentiate to some extent between different people in the given image, enabling them to be driven by multiple audio streams.

**Weaknesses**
* Regarding the setting, given a reference image with multi-stream audio (only 2 streams in the whole paper) and text, then generating the video. It seems that the correspondence between different audio streams and characters in the image still needs to be manually specified. Considering this, when obtaining the corresponding mask area, it is possible to achieve the same effect directly in the audio cross attention using the mask, and the model trained on single-person speaking data can also directly achieve this. Do authors consider this?
* For the Adaptive Person Localization, the premise for this to work stably is that there is no collapse in the generated video latents, which would result in a higher similarity. If there is a collapse, the similarity calculation may be affected, which might impact the final effect. Additionally, although the author claims that the Adaptive Person Localization with L-RoPE could handle situations with large movement range motion, neither the main paper nor the supplementary materials contain any related cases; most cases are left-right patterns. In that case, it can be achieved simply by using a mask on the audio cross-attention, as mentioned above.
* For the training strategy (Partial Parameter Multi-task). The fundamental reason may be that the distribution of the training data is relatively biased, possibly because the training data mostly consists of portrait videos, which do not include hands and bodies, leading to the phenomenon described by the author. Additionally, since only part of the weights have been trained, will there be a conflict between text control and audio control?

---

> ### Author Rebuttal · Authors · 2025-07-31
>
> We appreciate the reviewers’ positive feedback, including recognition of our introduction of a new multi-person task and the affirmation of the L-RoPE. Below, we address the specific concerns raised by the reviewers individually.
>
> ### Q1: The difference between L-RoPE and mask-based audio injection strategy.
>
> Thank you for your question regarding the correspondence between audio streams and persons, and the potential use of mask-based approaches.
> Our method does not require obtaining a mask for each frame of the generated video. Instead, we only need to establish the correspondence between each person with their audio stream in the reference image, which is a much simpler and more robust requirement.
>
> As you mentioned, the mask-based method is the same as in Figure 3c of our paper. The mask-based method explicitly binds a specific mask region to an audio stream, requiring a predefined mask for each frame throughout the video. However, this approach faces significant limitations: if the generated person moves outside the predefined mask area, the correspondence between speech and mouth motion can become unsynchronized, leading to degraded performance.
>
> In contrast, our proposed L-RoPE method leverages visual similarity to dynamically associate each human appearance with its corresponding audio input, without relying on fixed spatial masks for the whole video. This design enables strong generalization and robustness, especially when the generated characters undergo large motions or pose changes. As a result, L-RoPE maintains accurate audio-visual synchronization even in challenging scenarios, which is a key advantage over mask-based and single-person methods. We further validated the effectiveness of L-RoPE through additional experiments comparing these two strategies, as presented in the table, where the mask-based method corresponds to (c) and L-RoPE corresponds to (d). The proposed L-RoPE achieves superior results.
>
>
> |          | Sync-C↑ | Sync-D↓ |
> |:--------:|:-------:|:-------:|
> |     a    |   3.49  |  10.73  |
> |     b    |   3.07  |  11.26  |
> |     c    |   7.09  |   8.00  |
> | d (Ours) |   7.56  |   7.13  |
>
> ### Q2: Model collapse and large motion case.
>
> Thank you for your comments and suggestions regarding the stability of adaptive person localization.
> We agree that the stability of similarity-based localization depends on the quality of the underlying video foundation model. With a less robust model, there is indeed a risk of latent collapse. However, in our work, we use Wan2.1, which is currently one of the most stable open-source video generation models. In our extensive experiments, we have not encountered latent collapse or related issues.
>
> Regarding large motion, Figure 7 in our paper already demonstrates cases with significant movement. Additionally, we have conducted further tests involving both camera motion and substantial person movement, and our method is able to handle these scenarios effectively. We will include more such examples in the final version of the paper to better illustrate the robustness of our approach.
>
> ### Q3: Training strategy and conflict between different conditions.
>
> Thank you for your comments regarding the training strategy and potential conflicts in multi-task control.
> We agree that the distribution of the training data can affect the generation results. However, our training dataset does include a substantial number of samples featuring hands, bodies, and camera movement. Moreover, our findings are consistent with those reported in [1], which indicate that full parameter sharing in multi-task training can result in image quality degradation and distortions.
>
> Regarding the concern about potential conflicts between text control and audio control, we have conducted extensive testing and have not observed any conflicts. The model is able to follow text-based instructions effectively while maintaining accurate audio-driven lip synchronization.
>
> [1] OmniAvatar: Efficient Audio-Driven Avatar Video Generation with Adaptive Body Animation
>
> ### Q4: Generalize to more speakers and overlap scenarios.
>
> Thank you for your thoughtful questions regarding the generalization of the L-RoPE approach to scenarios with more than two speakers and overlap scenarios.
> Our L-RoPE framework demonstrates strong generalizability across varying numbers of speakers. In the two-speaker setting, we assign distinct, non-overlapping ranges of video and audio labels to each individual (e.g., video labels 0–4 for person 1 and 20–24 for person 2; audio labels 2 for person 1 and 22 for person 2). For scenarios involving three or more individuals, we conducted additional experiments to verify that this labeling scheme can be extended by assigning new, non-overlapping ranges (e.g., video labels 40–44 and audio label 42 for a third person). This flexible approach enables the model to accommodate a greater number of speakers simply by expanding the label ranges for both video and audio streams.
>
> Importantly, the L-RoPE extension strategy remains effective even when the number of persons during inference differs from that during training. By assigning dedicated label ranges, each person’s lip movements are accurately aligned with their respective audio streams. Furthermore, we observed that fine-tuning the model with data containing the same number of persons as inference can further improve the synchronization between audio and lip movements.
>
> For overlap scenarios, when individuals’ faces are visible—even if their bodies are partially occluded—our method consistently associates the audio stream with the correct person. In scenarios where a person’s face is fully occluded and not visible in the frame, the corresponding audio does not drive lip motion or appearance for the occluded individual. Notably, if a person is temporarily occluded and subsequently reappears, our model is able to accurately re-associate the audio stream with the correct individual upon their return.
>
> Visualization results for scenarios involving more speakers and overlap will be provided in the supplementary materials of the updated version.

---

> > ### Author Response · Authors · 2025-08-06
> >
> > Dear Reviewer,
> >
> > Thank you for your valuable comments and feedback. I have addressed all of your questions and suggestions in my previous responses. If there are any further questions or concerns, please feel free to let me know. I am happy to provide any additional clarification if needed.

---

> ### Comment · Reviewer_34Jy · 2025-08-06
> **Response to the rebuttal**
>
> Thanks for the authors' response. This resolves some of my doubts, but there are still the following questions.
>
> Q1: For the mask strategy, it is not necessary to provide it for every frame, as the task focuses on humans or characters, and tracking can be used to achieve this. The method proposed in this article still needs to establish an initial relationship between each character and their audios, which does not fundamentally differ much from providing a mask for the first frame; in addition, the advantage mentioned for large movements in characters, I have not seen any video demos.
>
> Q3: Opening full parameters training will lead to a decrease in image quality dose not make sense, this is most likely a problem with your training strategy and data quality; by the way, regarding the mentioned control conflict issue, let's assume a scenario: a character is putting something into their mouth with their hand (controlled by text), and they are also speaking (using audio). Assuming only the audio branch is trained, the text branch will also perform control. What is the reason there is no conflict in this situation if only train the audio branch?
>
> Given the above, I choose to keep my initial rating.

---

> > ### Author Response · Authors · 2025-08-06
> >
> > ## Q1: Mask Strategy and Tracking
> >
> > Thank you for your question regarding the use of tracking and mask strategies.
> >
> > While it is true that tracking can be used to obtain masks for each person in a video, tracking models require access to the full RGB video as input. However, our method focuses on audio-driven multi-person video generation, not video editing or dubbing. In our setting, the model does not have access to the full video before generation. The input to our method consists only of a single reference image, multiple audio streams, and a text prompt. **The video is the output of our model, not part of the input.** Therefore, it is not possible to perform tracking to obtain masks for non-existent frames prior to generation.
> >
> > To address this challenge, we propose the Adaptive Person Localization method, which utilizes the attention map and relies solely on the mask from the reference image. During the denoising process, this approach leverages visual similarity to estimate the motion regions of each person, thereby eliminating the need for tracking. As a result, our method enables robust person-audio association even in the absence of an input video.
> >
> > ## Q2: Training Strategy
> >
> > Thank you for your comment regarding the impact of full parameter training on image quality.
> >
> > We acknowledge that the decrease in image quality observed with full parameter training may be attributable to differences in data distribution between our training set and the Wan2.1 base model, as well as limitations in our training data and batch size imposed by computational constraints. However, we emphasize that both full parameter training and self-attention (cross-attention) training in our experiments were conducted using identical datasets and under consistent experimental conditions. The degradation in image quality—particularly in challenging regions such as the hands—is evident.
> >
> > It is important to note that this phenomenon is not unique to our work. Similar observations have been reported in other studies, such as OmniAvatar [1], which also found that full parameter training can result in image quality degradation and distortions, especially in human motion elements, like hands and mouths
> >
> > [1] OmniAvatar: Efficient Audio-Driven Avatar Video Generation with Adaptive Body Animation
> >
> > ## Q3: Conflict Issue
> >
> > Thank you for raising this interesting scenario regarding potential control conflicts.
> >
> > In our current framework, there is generally no conflict between audio-driven and text-driven controls when the actions involve separate body parts—for example, when a character is picking up an object (controlled by text) while simultaneously speaking (controlled by audio), as these controls operate independently. However, when the actions overlap—such as when a character is putting something into their mouth (text control) and speaking at the same time (audio control)—a conflict arises. In this case, the mouth cannot be driven by the audio stream. This is a limitation shared by all current audio-driven human animation methods, not just ours. Addressing such complex scenarios would require a new paradigm; for example, joint audio-visual generation methods may offer a promising solution.

---

> > > ### Comment · Reviewer_34Jy · 2025-08-08
> > > **Response to the rebuttal**
> > >
> > > Thanks for the author's reply, which has addressed some of my concerns. Finally, I suggest that the author provide more examples of people moving and scenes with more than two persons, as these are sufficient to prove what the paper claims. Currently, due to the lack of these examples, I maintain my rating, but I am not opposed to accepting this paper.

---

### Official Review · Reviewer_M1mM · 2025-07-02

**Clarity:** 3
**Significance:** 3
**Originality:** 3
**Rating:** 5
**Confidence:** 5

**Summary:**

The paper proposed a new task (audio-driven multi-person conversational video generation), a framework MultiTalk for this task employing L-RoPE introduced for multi-stream audio injection. The MultiTalk is evaluated in generating various type of video generation such as talking face, talking body, and multi-person conversation.

**Questions:**

The paper collects a new dataset for multi-person conversation, but the paper does not mention the dataset is proposed. Why authors do not propose the dataset?

The time to require to generate videos is not explored at all. Since it is not the main aim and not in the scope, it might be understandable. However, if the generation is computationally expensive and requires time (most probably and as expected), I suggest to mention this in the limitation at least and evaluate the model’s response time. Can you include elapsed time to generate videos?

The paper investigates different injection strategies for multi-stream audio. An ablation study can be included to see the impact of injection strategies, otherwise the investigation of injection strategies will be incomplete. Can you make such ablation study?
Why authors won’t share the collected dataset?

**Ethical Concerns:**

["NO or VERY MINOR ethics concerns only"]

**Final Justification:**

My concerns are addressed, so I decided to increase my score to Accept from Borderline Accept.

**Limitations:**

yes

**Quality:**

3

**Strengths And Weaknesses:**

Strengths:

1. The writing is easy to follow and supported by equations, figures, details of implementation and evaluation well.

2. The model has better scores than SOTA models in terms of the video quality, facial expressiveness, and lip synchronization.

3. The model can also generate multi-person and interactive videos well.

Weaknesses:

1. The definition of the proposed new task (audio-driven multi-person conversational video generation) is brief. It seems incremental as existing talking face models (that are trained to generate talking face videos with a single face) can generate multi face by generating talking face videos independently then combine videos. So, the paper can include the formal definition of the new task in the coloration with existing tasks.

2. The model supports multi-person conversation, but it supports dual-person only. This can be mentioned in the limitation.

3. The proposed model cannot preserve the identity of the persons in the reference image. See the second singing video in the supplementary materials (the male character becomes someone else).

Checklist related:

1. Checklist 4. Although authors provide the details of the experiments, the collected dataset won't be shared, so it questions the reproducibility of the experiments.

2. Checklist 10. The broader impact is not limited to the generating fake videos of celebrities. Please elaborate it. Moreover, I suggest putting the broader impact in the main manuscript instead of supplementary material because it is vital ethical concern of the talking face models.

---

> ### Author Rebuttal · Authors · 2025-07-31
>
> We appreciate the reviewers’ positive feedback, including recognition of our clear and detailed writing, superior generation results, and strong capabilities in multi-person and interactive video generation. Below, we address the specific concerns raised by the reviewers individually.
>
>
>
>
>
> ### Q1: Definition of the new task and its distinction from single-person talking face task.
>
> Thank you for your valuable feedback. We appreciate your suggestion to provide a more formal definition of our proposed task and to clarify its distinction from existing single-person talking face generation tasks.
>
>
> Audio-driven multi-person conversational video generation is defined as follows: Given a reference image containing multiple persons and corresponding audio streams (with a one-to-one correspondence between each person and their audio), the goal is to synthesize a video sequence in which all persons appear together in the same frame, and each person’s lip movements are temporally synchronized with their respective audio input. Unlike previous single-person talking face generation tasks, this new task requires the joint modeling of multi-person interactions, spatial consistency, and audio-visual synchronization within a unified, end-to-end generative framework, requiring only a single diffusion process.
>
> In contrast to approaches that generate videos for each person independently and subsequently composite them, this new task demands integrated modeling of multiple individuals, offering several key advantages:
>
> 1. Higher computational efficiency: Only a single inference process is required, substantially reducing computational costs.
>
> 2. Global consistency: The unified framework enables better control over the overall coherence of the generated content, such as coordinated camera movements, lighting, and scene dynamics.
>
> 3. Enhanced interaction modeling: This approach is inherently more suitable for capturing interactions among individuals, enabling natural and contextually appropriate reactions (e.g., when one person is speaking, others can display attentive or responsive behaviors).
>
> We will revise the paper to include a formal definition of the new task and add a dedicated section to compare it with existing single-person talking face tasks. This will help clarify the novelty and technical challenges of our work.
>
>
>
>
> ### Q2: Generalize to more speakers.
>
> Thank you for your thoughtful questions regarding the generalization of the L-RoPE approach to scenarios with more than two speakers and varying numbers of persons at inference time.
> Our L-RoPE framework demonstrates strong generalizability across varying numbers of speakers. In the two-speaker setting, we assign distinct, non-overlapping ranges of video and audio labels to each individual (e.g., video labels 0–4 for person 1 and 20–24 for person 2; audio labels 2 for person 1 and 22 for person 2). For scenarios involving three or more individuals, we conducted additional experiments to verify that this labeling scheme can be extended by assigning new, non-overlapping ranges (e.g., video labels 40–44 and audio label 42 for a third person). This flexible approach enables the model to accommodate a greater number of speakers simply by expanding the label ranges for both video and audio streams.
>
> Importantly, the L-RoPE extension strategy remains effective even when the number of persons during inference differs from that during training. By assigning dedicated label ranges, each person’s lip movements are accurately aligned with their respective audio streams. Furthermore, we observed that fine-tuning the model with data containing the same number of persons as inference can further improve the synchronization between audio and lip movements.
>
> Visualization results for scenarios involving more speakers will be provided in the supplementary materials of the updated version.
>
> ### Q3: Identity degradation in the generated videos.
>
> Thank you for pointing out the issue regarding identity preservation in our generated videos.
> We acknowledge that identity degradation may occur in generated videos, particularly when the video duration is excessively long or when there are substantial changes in facial motion angles. This limitation arises because the video foundation model we employ also suffers from the identity degradation problem. On one hand, as video foundation models continue to advance, the issue of identity degradation is expected to be mitigated. On the other hand, in future work, we plan to investigate explicit identity injection techniques to further enhance identity preservation.
>
> Thank you again for your valuable feedback.
>
>
>
> ### Q4: Dataset release and experimental reproducibility.
>
> Thank you for your comments and concerns regarding the dataset and reproducibility.
> All data used in our experiments were collected from publicly available sources on the internet. Our data collection process follows the best practices established by previous works [1]-[3].
> For data preprocessing, we closely follow the procedures described in [1] and further filter out samples exhibiting large facial movements or unsynchronized speech and mouth motion [4], and do not have any unique data processing procedures.
>
> We acknowledge the importance of reproducibility and are committed to supporting the community.
> We will release the MTHM test dataset, along with the corresponding code and model weights, to the research community to encourage reproducibility and facilitate further research and development in audio-driven human animation.
>
>
> [1] Koala-36M: A Large-scale Video Dataset Improving Consistency between Fine-grained Conditions and Video Content
>
> [2] Openvid-1m: A large-scale high-quality dataset for text-to-video generation
>
> [3] MiraData: A Large-Scale Video Dataset with Long Durations and Structured Captions
>
> [4] Out of time: automated lip sync in the wild
>
> ### Q5: Broader impact of the paper.
>
> Thank you for highlighting the importance of discussing the broader impact and ethical considerations of our work. This paper introduces an effective approach for audio-driven multi-person conversational video generation to the community. However, this technology also raises significant ethical concerns. Beyond the risk of generating fake videos of celebrities, there are broader implications, including the potential for misuse in creating deepfakes for misinformation, defamation, fraud, or harassment. Such synthetic videos could be used to impersonate individuals, manipulate public opinion, or violate privacy. These risks are not unique to our approach but are shared across the broader field of human animation and generative media.
>
> Thank you for your suggestion. We will include the discussion of broader impact in the main manuscript in the updated version.
>
>
>
>
>
> ### Q6: Inference time.
>
> Thank you for your valuable suggestion regarding the evaluation of video generation time.
> Although our method introduces an additional audio condition, the computational time required to pass through the DiT backbone remains the same as in Wan2.1. On an NVIDIA A100 GPU, each inference step in our method takes approximately 35 seconds, resulting in a total generation time of about 1,430 seconds for a 40-step inference. Furthermore, all acceleration strategies available for Wan2.1—such as TeaCache and model distillation—are also applicable to our approach. For example, when employing a distilled model (such as lightx2v), the total inference time is reduced to approximately 170 seconds per video.
>
> ### Q7: Audio injection results.
>
> Thank you for the valuable comments.
> We conducted an additional ablation study to investigate the impact of different audio injection strategies. The results are summarized in the table below, with each row corresponding to an audio injection strategy as illustrated in Figure 3. Strategies (a) and (b) fail to bind multi-stream audio to the corresponding video latent regions. Strategy (c) employs a hard mask-based audio binding approach, which is capable of associating multi-stream audio with different persons; however, its effectiveness is limited to videos with minimal motion. When a person exhibits extensive movement, this strategy also results in failure cases. In contrast, our proposed L-RoPE method (d) achieves the best results across all tested scenarios, demonstrating the superiority of our approach. We will provide these results in the supplementary materials of the updated version.
>
>
>
> |          | Sync-C↑ | Sync-D↓ |
> |:--------:|:-------:|:-------:|
> |     a    |   3.49  |  10.73  |
> |     b    |   3.07  |  11.26  |
> |     c    |   7.09  |   8.00  |
> | d (Ours) |   7.56  |   7.13  |

---

> > ### Comment · Reviewer_M1mM · 2025-08-05
> > **Reply**
> >
> > Thank you for the additional effort in addressing my concerns. I appreciate the inclusion of a formal definition of the task, new results demonstrating the model’s ability to generalize to more speakers, elaboration on identity degradation, commitment to releasing the dataset upon acceptance (which will be valuable to the community), expanded discussion on broader impact, inference time analysis, and the ablation study.
> >
> > Since my concerns have been addressed, I have decided to increase my rating to Accept.
> >
> > However, I have also read the other reviews, and I would like to share my thoughts on the ethical concerns raised, particularly regarding data collection:
> >
> > Use of YouTube videos: The use of publicly available YouTube videos for research purposes may fall under the Fair Use doctrine, even without explicit permission. I think this is a reasonable interpretation given the academic context of the work.
> >
> > Bias in the dataset: Gender and racial bias are clearly present in the collected dataset. While this is a significant ethical concern, the lack of attention to these biases can be seen as a limitation that diminishes the value of the dataset. It is important to acknowledge when the dataset is published and address such biases in future work.
> >
> > Since the paper offers substantial contributions beyond the dataset itself, I believe that acknowledging these ethical concerns—specifically the Fair Use basis for data collection and the presence of bias—should be sufficient.

---

> > > ### Author Response · Authors · 2025-08-05
> > >
> > > Thank you very much for your thoughtful and constructive feedback, as well as your recognition of our efforts to address your concerns. We greatly appreciate your positive evaluation and recommendation for acceptance.
> > >
> > > We also thank you for sharing the ethical considerations regarding data collection and dataset bias. Regarding the EMTD test dataset, we will release it to the community upon acceptance of the paper. Concerning the issue of bias, we are committed to addressing and mitigating such concerns in our future work. Wishing you continued success in your work and good health.

---

### Official Review · Reviewer_cNEL · 2025-07-03

**Clarity:** 3
**Significance:** 3
**Originality:** 3
**Rating:** 5
**Confidence:** 3

**Summary:**

The paper introduces a new task: audio-driven multi-person conversational video generation. It proposes a new framework for this task, incorporating a Label Rotary Position Embedding method (L-RoPE) to address the problem of inaccurate audio binding. Additionally, the paper investigates several schemes for multi-stream audio injection and explores a set of training strategies, including two-stage training, partial parameter training, and multi-task training.

**Questions:**

Please clarify the contents in weakness.

**Ethical Concerns:**

["NO or VERY MINOR ethics concerns only"]

**Limitations:**

yes

**Quality:**

3

**Strengths And Weaknesses:**

Strengths:
1. The paper introduces an interesting task: audio-driven multi-person conversational video generation. The task is also challenging due to multi-stream audio input, the audio binding problem, and person localization in video.
2. The proposed framework, including audio-video cross-attention and audio-person binding, is reasonable, and the writing is easy to follow.
3. The video results for multiple persons are promising.

Weaknesses:
1. The paper investigates several schemes for multi-stream audio injection. Showing the results of these alternative schemes could further strengthen the advantages of L-RoPE.
2. The proposed task is interesting and inspiring. This may be a minor issue, but since the task focuses on multi-person scenarios, it would be beneficial to discuss how the framework could be extended to more than two persons and the challenges involved in scaling to additional persons.

---

> ### Author Rebuttal · Authors · 2025-07-31
>
> We appreciate the reviewers’ positive feedback, including recognition of our introduction of a new multi-person task, the reasonableness of our method, the clarity of our writing, and the promising results. Below, we address the specific concerns raised by the reviewers individually.
>
>
> ### Q1: Ablation study for multi-stream audio injection.
>
> Thank you for the valuable comments.
> We conducted an additional ablation study to investigate the impact of different audio injection strategies. The results are summarized in the table below, with each row corresponding to an audio injection strategy as illustrated in Figure 3. Strategies (a) and (b) fail to bind multi-stream audio to the corresponding video latent regions. Strategy (c) employs a hard mask-based audio binding approach, which is capable of associating multi-stream audio with different persons; however, its effectiveness is limited to videos with minimal motion. When a person exhibits extensive movement, this strategy also results in failure cases. In contrast, our proposed L-RoPE method (d) achieves the best results across all tested scenarios, demonstrating the superiority of our approach. We will provide these results in the supplementary materials of the updated version.
>
>
> |          | Sync-C↑ | Sync-D↓ |
> |:--------:|:-------:|:-------:|
> |     a    |   3.49  |  10.73  |
> |     b    |   3.07  |  11.26  |
> |     c    |   7.09  |   8.00  |
> | d (Ours) |   7.56  |   7.13  |
>
> ### Q2: Generalize to more speakers.
>
> Thank you for your thoughtful questions regarding the generalization of the L-RoPE approach to scenarios with more than two speakers and varying numbers of persons at inference time.
> Our L-RoPE framework demonstrates strong generalizability across varying numbers of speakers. In the two-speaker setting, we assign distinct, non-overlapping ranges of video and audio labels to each individual (e.g., video labels 0–4 for person 1 and 20–24 for person 2; audio labels 2 for person 1 and 22 for person 2). For scenarios involving three or more individuals, we conducted additional experiments to verify that this labeling scheme can be extended by assigning new, non-overlapping ranges (e.g., video labels 40–44 and audio label 42 for a third person). This flexible approach enables the model to accommodate a greater number of speakers simply by expanding the label ranges for both video and audio streams.
>
> Importantly, the L-RoPE extension strategy remains effective even when the number of persons during inference differs from that during training. By assigning dedicated label ranges, each person’s lip movements are accurately aligned with their respective audio streams. Furthermore, we observed that fine-tuning the model with data containing the same number of persons as inference can further improve the synchronization between audio and lip movements.
>
> Visualization results for scenarios involving more speakers will be provided in the supplementary materials of the updated version.

---

> > ### Comment · Reviewer_cNEL · 2025-08-06
> >
> > Thank you for the additional results and clarification. Please update the results in the paper. I will vote for accept as my rating.
> >
> > In additional, I read the ethical concerns from the other reviewers. Please remember to clarify the data resource and license in the paper.

---

### Official Review · Reviewer_reUP · 2025-07-05

**Clarity:** 4
**Significance:** 4
**Originality:** 3
**Rating:** 5
**Confidence:** 4

**Summary:**

This paper proposes MultiTalk, a framework for generating multi-person conversational videos driven by multi-stream audio and prompts. The authors tackle the problem of binding each audio stream to the correct person using a novel Label Rotary Position Embedding. They also design partial parameter training and multi-task learning to preserve instruction-following ability. Experiments on talking head, body, and multi-person datasets show improved synchronization and video quality over prior single-person methods.

**Questions:**

- The instruction-following claims are promising, but current prompts in your experiments are relatively simple. How does the model perform when facing complex, compositional prompts (e.g., involving emotions, multiple actions, overlapping speech, or indirect commands)? Can you provide such examples or failure cases?

- The current experiments focus on two-person scenarios. Does the L-RoPE approach generalize to cases with 3 or more speakers? How does the model handle varying numbers of persons at inference time, especially when the number of audio streams and persons may differ from training?

- The paper mentions a performance gap between real and synthetic audio. Could you elaborate on how significant this gap is, and whether your model could be made robust to TTS-generated speech, which is common in practical applications?

**Ethical Concerns:**

["NO or VERY MINOR ethics concerns only"]

**Final Justification:**

The authors have satisfactorily addressed my pre-rebuttal concerns. They demonstrated generalization of the L-RoPE approach to more than two speakers and to scenarios with variable numbers of persons at inference time, with a clear labeling strategy and supporting experiments. They clarified the performance gap between real and TTS-generated audio and proposed concrete strategies to improve robustness.

Given the novelty of the multi-person conversational video generation task, the impressive qualitative results, and the thorough clarifications provided, I maintain my positive assessment and recommendation for acceptance.

**Limitations:**

Yes.

**Paper Formatting Concerns:**

None.

**Quality:**

4

**Strengths And Weaknesses:**

Strengths:
- Proposes the new task of multi-person conversational video generation, extending beyond prior single-person animation works.

- The qualitative results shown are impressive. The generated videos demonstrate highly realistic lip-sync, body motion, and person-to-audio alignment, making the visual results very impressive.

- The introduction of the Label Rotary Position Embedding to solve the key challenge of binding each person to their correct audio stream is both novel and effective.

-  The paper provides thorough experimental validation, including detailed ablations on multi-stream audio injection methods and training strategies. The paper also carefully studies how partial parameter tuning and multi-task learning contribute to maintaining performance with limited compute and data.

- The method is validated across multiple datasets and scenarios (talking head, talking body, multi-person), showing consistently better synchronization and video quality than prior methods.

Weaknesses:
- Although the adaptive person localization is mentioned, it is somewhat unclear how robust this method is in highly occluded or overlapping person scenarios.

- The paper claims instruction-following ability, but most prompts in experiments are relatively simple (“person holds cup,” “person hugs”). It is unclear how well the model handles more abstract or compositional instructions like multi-step narratives, emotional expressions, or simultaneous actions.

---

> ### Author Rebuttal · Authors · 2025-07-31
>
> We appreciate the reviewers’ positive feedback, including recognition of our introduction of a new multi-person task, impressive qualitative results, the novelty and effectiveness of our method, thorough experimental validation, and superior performance. Below, we address the specific concerns raised by the reviewers individually.
>
>
> ### Q1: Robustness under occlusion and overlap scenarios.
>
> Thank you for your question regarding the robustness of our adaptive person localization method in highly occluded or overlapping scenarios.
> When individuals’ faces are visible—even if their bodies are partially occluded—our method consistently associates the audio stream with the correct person. In scenarios where a person’s face is fully occluded and not visible in the frame, the corresponding audio does not drive lip motion or appearance for the occluded individual. Notably, if a person is temporarily occluded and subsequently reappears, our model is able to accurately re-associate the audio stream with the correct individual upon their return. The corresponding visualization results will be provided in the supplementary materials of the updated version.
>
>
>
> ### Q2: Handling more complex instructions.
>
> Thank you for your insightful comment regarding the instruction-following capabilities of our model.
> We conducted additional evaluations to assess the robustness of our method on more complex and compositional prompts:
>
>
> **Multi-step narratives**: Since Wan2.1 can only generate 81 frames per chunk, it is challenging to complete multiple actions within a single chunk. Hence, we decompose multi-step narratives into several actions, generating each chunk with prompts specifying a few actions. Through leveraging the long video generation mentioned in the paper, our model is able to represent multi-step narratives effectively.
>
> **Emotional expressions**: We use the same reference image, and specify emotional (e.g., angry, sad, happy) via the text prompt. Our model successfully generates videos with the corresponding emotional expressions.
>
> **Simultaneous actions**: In multi-person scenarios, we assign different actions to each person via text prompts. The resulting videos demonstrate that both individuals can perform their respective actions simultaneously.
>
> **Overlapping speech**: By providing separate audio streams for each person, our model generates synchronized lip movements for both speakers without interference, thus achieving overlapping speech.
>
> **Indirect commands**: Our model currently struggles with indirect commands, likely due to the base video model’s limited ability to understand such instructions.
>
> Due to rebuttal format constraints, we are unable to include success and failure visualization results here. However, we will provide these examples in the updated version.
>
> ### Q3: Generalization to more speakers and variable person numbers.
>
> Thank you for your thoughtful questions regarding the generalization of the L-RoPE approach to scenarios with more than two speakers and varying numbers of persons at inference time.
> Our L-RoPE framework demonstrates strong generalizability across varying numbers of speakers. In the two-speaker setting, we assign distinct, non-overlapping ranges of video and audio labels to each individual (e.g., video labels 0–4 for person 1 and 20–24 for person 2; audio labels 2 for person 1 and 22 for person 2). For scenarios involving three or more individuals, we conducted additional experiments to verify that this labeling scheme can be extended by assigning new, non-overlapping ranges (e.g., video labels 40–44 and audio label 42 for a third person). This flexible approach enables the model to accommodate a greater number of speakers simply by expanding the label ranges for both video and audio streams.
>
> Importantly, the L-RoPE extension strategy remains effective even when the number of persons during inference differs from that during training. By assigning dedicated label ranges, each person’s lip movements are accurately aligned with their respective audio streams. Furthermore, we observed that fine-tuning the model with data containing the same number of persons as inference can further improve the synchronization between audio and lip movements.
>
> Visualization results for scenarios involving more speakers will be provided in the supplementary materials of the updated version.
>
>
>
> ### Q4: Performance gap between real and synthetic audio.
>
> Thank you for your valuable question regarding the performance gap between real and synthetic (TTS-generated) audio.
> The observed gap can be primarily attributed to a domain discrepancy: our model is trained exclusively on real audio, which typically contains rich emotional cues and natural prosody. As a result, the generated videos exhibit more expressive and realistic facial behaviors when driven by real audio. In contrast, most current TTS-generated audio lacks emotional variation and nuanced expressiveness, leading to video outputs that appear less vivid and natural. A promising approach to address this issue is to augment the training data with TTS-generated audio, thereby enabling the model to better generalize to synthetic audio inputs. More detailed analysis and experimental results on TTS robustness will be provided in the updated version.

---

> > ### Comment · Reviewer_reUP · 2025-08-07
> >
> > Thank you for the detailed rebuttal. It effectively addressed most of my concerns. I will maintain my positive score.

---

### Comment · Area_Chair_Mi7E · 2025-08-01
**Reviewer-author discussion period**

Dear reviewers and authors,

We are now in the reviewer-author discussion period until Aug 6 11:59pm AoE. The authors have posted a detailed response to each review. At your earliest convenience, please read all the reviews and rebuttals, and respond as soon as you can to the author's rebuttals to start discussion. At minimum, please respond to the author's rebuttal to indicate you have read it. The discussion period is not very long, so it would be good to ensure there is time for back-and-forth if needed. Here are some suggested points of discussion:

- Reviewer s1Za may have had some misunderstandings about the paper in terms of relation to prior work and evaluation/ablation results; could you please have a look at the authors' rebuttal and see if it helps resolve these for you?

- Reviewer 34Jy raised some concerns about latent collapse and training data distribution assumptions; could you please read the authors' rebuttal to see if they help clarify these issues for you?

- There is some general concern about the dataset used in the paper, but the authors have provided additional information about this. Does this additional information assuage your concerns? Authors, please also respond to the ethics reviewer's specific questions.

Thanks for all your efforts.

Best, AC

---

### Note · Authors · 2025-08-13

We would like to sincerely thank the reviewers and the area chair for their thoughtful feedback and constructive suggestions throughout the review process. We have carefully addressed the comments and concerns raised during the rebuttal and discussion phases, and appreciate the opportunity to clarify our contributions and methodology.

In particular, we have provided detailed responses regarding the generalization to occlusion and overlap scenarios, handling more complex instructions, extension to interactions involving more than two persons, the performance gap between real and synthetic audio, quantitative comparisons of different audio injection methods, the definition of our proposed new task, explanation of the identity degradation issue, the broader impact of this technology, inference time analysis, comparison between L-RoPE and mask-based audio injection methods, model stability, clarification of conflicting conditions, quantitative comparisons of different training strategies, clarification of our evaluation protocols, explanation of Table 3, and clarification of our data collection process. We believe these clarifications further demonstrate the novelty and significance of our work.

We would like to reiterate that our data collection and usage practices strictly adhere to community standards and ethical guidelines, as discussed in our previous responses. In particular, we appreciate Reviewer M1mM’s thoughtful discussion of ethical considerations, including the recognition that the use of publicly available YouTube videos for academic research can be reasonably interpreted under the Fair Use doctrine. Given that most of our training data is in English and language is strongly correlated with race, we acknowledge the importance of transparency regarding potential gender and racial biases in the dataset, as highlighted by the reviewers, and we will explicitly address these limitations in future work. Our methodology and experiments are consistent with prior influential works in the field and recent NeurIPS-accepted papers, and we respectfully request that our work be evaluated under the same standards and with equal consideration.

Once again, we thank the reviewers and the area chair for their time and effort in evaluating our submission. We hope our responses have addressed all outstanding concerns, and respectfully ask for your consideration of our work.

---

### Decision · Program_Chairs · 2025-09-17

**Decision:**

Accept (poster)

**Comment:**

"This paper approaches the task of multi-person audio-to-human generation using video Diffusion Transfomers. It specifically aims to address the issues of having multiple people present in the output video, which previous methods do not handle well. This work proposes the use of Label Rotary Position Embedding (L-RoPE) which helps to bind audio streams to the correct tokens (and therefore regions) in the diffusion process."

Strengths
- The paper tackles a challenging problem: generation of multi-person conversational videos driven by multi-stream audio and prompts.

- The paper provides a novel mechanism, label rotary position embedding (L-RoPE) to address the issue to resolve binding between audio speakers and visual persons.

- As mentioned and inquired by Reviewers cNEL and M1mM, the approach can be generalized beyond 2 speakers (as stated by authors in rebuttal)

Weaknesses
- Reviewers (particularly s1Za) raised ethical concerns about the dataset collected by the paper in terms of privacy, consent, copyright and fair use, and representation/bias. The ethics reviewer investigated this, and requested information from the authors on source of the training dataset collected from YouTube, the license of the data, and consent of the subjects. The authors provided the required information in the comments, and the paper seems to follow similar practices for uses of YouTube data of recent NeurIPS 2023/204 papers (as noted by the ethics reviewer). Thus, I think the issue of the dataset is not a rejection factor for this paper, but the authors MUST provide all the information requested by the ethics reviewer, apply the suggested mitigations for

- Paper lacks reproducibility currently, but authors promise to release the test dataset, code, and model weights.

- More demos of moving people and more than 2 speakers are necessary.

Decision

This paper provides an effective and novel approach for the difficult problem of multi-person audio-drive conversations. Concerns were raised about the training dataset collection, but the authors have provided information requested by ethics reviewer, and this information MUST be included in the final version. During rebuttal, a number of other issues were raised by reviewers, which the authors provided satisfactory responses to, causing reviewers to raise their score. These concerns, as summarized by the authors, are: "generalization to occlusion and overlap scenarios, handling more complex instructions, extension to interactions involving more than two persons, the performance gap between real and synthetic audio, quantitative comparisons of different audio injection methods, the definition of our proposed new task, explanation of the identity degradation issue, the broader impact of this technology, inference time analysis, comparison between L-RoPE and mask-based audio injection methods, model stability, clarification of conflicting conditions, quantitative comparisons of different training strategies, clarification of our evaluation protocols, explanation of Table 3, and clarification of our data collection process."